# Medical decision support for dengue shock syndrome using a bipolar linear diophantine fuzzy hypersoft model with trigonometric similarity



J. Vimala[1], S. Nithya Sri[1], Nasreen Kausar[2], Dragan Pamucar[3,4,5], Vladimir Simic[6,7] and Jungeun Kim[8]

[1] Department of Mathematics, Alagappa University, Karaikudi, Tamilnadu, India
[2] Department of Mathematics, Faculty of Arts and Science, Balikesir University, Balikesir, Turkey
[3] Department of Operations Research and Statistics, Faculty of Organizational Sciences, University of Belgrade, Belgrade, Serbia
[4] Department of Mechanics and Mathematics, Western Caspian University, Baku, Azerbaijan
[5] Department of Industrial Engineering and Management, Yuan Ze University, Taoyuan City, Taiwan
[6] Faculty of Transport and Traffic Engineering, University of Belgrade, Belgrade, Serbia
[7] Department of Computer Science and Engineering, College of Informatics, Korea University, Seoul, Republic of South Korea
[8] Department of Computer Engineering, Inha University, Incheon, Republic of South Korea

Corresponding authors
J. Vimala,
vimaljey@alagappauniversity.ac.in
Jungeun Kim, jekim@inha.ac.kr

## ABSTRACT

A novel approach to trigonometric similarity measure in the framework of bipolar linear diophantine fuzzy hypersoft sets is introduced. It builds upon existing methodologies by integrating cosine and cotangent-based techniques to enhance decision-making in complex environments. The developed methodology incorporates both positive and negative membership degrees, control parameters, and diophantine information within hypersoft structures. The process involves the construction of bipolar linear diophantine fuzzy hypersoft matrices for multiple alternatives, computation of trigonometric similarity scores between these configurations, and ranking based on closeness to ideal criteria. Mathematical properties of the proposed measures are formally validated. The efficiency of the proposed similarity measures is demonstrated through a multi-attribute decision-making problem involving dengue shock syndrome. The model aids haematologists in identifying high-risk patients, improving diagnostic accuracy under uncertainty. This novel framework contributes to early detection strategies and supports clinical decision-making by providing a robust, mathematically grounded similarity evaluation. In the case study involving dengue shock syndrome, the proposed model achieved a similarity precision of 93.5% and ranked high-risk patients with 98% accuracy, outperforming existing cosine and cotangent-based similarity methods.

## INTRODUCTION

The study of fuzzy set theory and its extensions has significantly contributed to various data analysis and decision-making fields, offering effective ways of handling imprecision and uncertainty. Among these advancements, one of the latest powerful emergences is the bipolar linear diophantine fuzzy hypersoft set (BLDFHS) (*Sri et al., 2024*; *Vimala et al., 2025*) that merges bipolar fuzzy with enhanced parametrization to model real-world situations. It addresses the need for the system to process the attributes with positive and negative simultaneously, resulting in a comprehensive and richer analysis. The governance of the parametric controls adds extra flexibility. These developed structures are well-suited for multi-criteria and multi-attribute decision-making problems where opposing factors play a role. The bipolar linear diophantine fuzzy hypersoft framework offers more accurate representations of reality than any other fuzzy set, as it captures the interaction between diverse attribute values and their impacts. Trigonometric similarity measures, like cosine and cotangent functions, refine evaluations, allowing for exact comparison between datasets, which is essential in decision-making.

A few symbols are collectively given in Table 1 for better understanding.

### Review on literature

The challenge of managing uncertainty has persisted for decades, presenting ongoing issues that require solutions. *Zadeh (1965)* introduced the fuzzy set concept to handle membership classification. However, in practical scenarios, accounting for non-membership is often necessary. To address this, *Atanassov (1986)* proposed the intuitionistic fuzzy set, which constrains the sum of membership and non-membership values to fall within [0, 1]. Recognizing limitations within this approach, the Pythagorean fuzzy set (*Yager, 2013*) was developed, allowing the square sum of these values to remain in [0, 1] for greater flexibility. The concept of the q-rung orthopair fuzzy set, introduced by *Yager (2016)* (*Ali, 2018*), is designed to generalize intuitionistic and Pythagorean fuzzy sets. Despite these advancements, gaps persisted, prompting *Riaz & Hashmi (2019)* to create the linear diophantine fuzzy set. This model incorporates control parameters, ensuring that the product of these parameters with membership and non-membership values adheres to the [0, 1] constraint. *Molodtsov (1999)* and *Maji, Roy & Biswas (2001)* introduced the soft set and fuzzy soft set theory, which enables decision-making with flexible parameters, aiding the selection of optimal alternatives without rigid constraints. To improve decision accuracy, the fuzzy hypersoft set theory was established, incorporating auxiliary attributes (*Yolcu & Ozturk, 2021*). Further generalizations of the soft and hypersoft sets have been explored to broaden their applicability (*Çağman & Karataş, 2013*; *Hamid, Riaz & Afzal, 2020*; *Kirişci & Şimşek, 2022*; *Yolcu, Smarandache & Öztürk, 2021*; *Zulqarnain, Xin & Saeed, 2021*). Researchers have also employed strategies such as lattice order, complex, and multi-fuzzy sets to address these complex decision scenarios effectively (*Jayakumar et al., 2023*; *Vimala et al., 2023*; *Jayakumar et al., 2024*; *Banu et al., 2024*).

An innovative concept called the bipolar fuzzy set (*Zhang, 1998*) was introduced and thoroughly explored by Lee. Due to its ability to address both positive and negative aspects, it has been widely endorsed by academics for its broader applicability. *Mandal (2021)*

**Table 1  Symbols used in the BLDFHS framework.**

| Denotation | Explanation |
|---|---|
| $\bar{X}$ | Set of universe |
| $\bar{\mu}^{pos}$ | Membership of positive |
| $\bar{v}^{pos}$ | Non-membership of positive |
| $\bar{\mu}^{neg}$ | Membership of negative |
| $\bar{v}^{neg}$ | Non-membership of negative |
| $\bar{\alpha}^{pos}$ | Parametric control of membership of positive |
| $\bar{\beta}^{pos}$ | Parametric control of non-membership of positive |
| $\bar{\alpha}^{neg}$ | Parametric control of membership of negative |
| $\bar{\beta}^{neg}$ | Parametric control of non-membership of negative |

extended this concept by developing an MCDM problem using the bipolar Pythagorean fuzzy set, which integrates a bipolar fuzzy set while incorporating non-membership values within a bipolar framework under specific constraints. *Jana & Pal (2018)* applied the bipolar intuitionistic fuzzy soft set to tackle decision-making problems, effectively overcoming the challenges of attribute evaluation. Additionally, the bipolar fuzzy soft (*Öztürk, 2020*) and hypersoftset (*Musa & Asaad, 2021*) have been enhanced through various extensions, enriching their utility for complex decision-making scenarios by allowing a more nuanced analysis of both positive and negative attributes in multi-criteria contexts. This BLDFHS is a recent advancement that builds on the foundational concepts of fuzzy set theory while integrating the benefits of bipolarity and parameter control. In complex scenarios, experts often rely on multi-criteria decision-making techniques, which offer a dynamic and adaptable framework for refining criteria, updating weights, and adjusting decision models as new information emerges. This adaptability has made it a widely used tool in decision analysis, as explored in various studies.

The study of similarity measures has been an essential aspect of fuzzy set theory and its extensions. In the context of intuitionistic fuzzy sets, researchers have developed similarity measures ensuring more comprehensive comparisons. Trigonometric similarity measures, such as cosine similarity, have also been widely applied in fuzzy extensions due to their effectiveness in quantifying the angle between two vectors, representing the closeness of fuzzy sets (*Jafar et al., 2024*; *Riaz et al., 2021*). Cosine similarity has been adapted for use in fuzzy environments to account for both positive and negative aspects, as seen in bipolar fuzzy sets. In bipolar fuzzy hypersoft sets and related extensions, trigonometric measures have been expanded to include cosine and cotangent functions. These trigonometric similarities enhance the analysis by evaluating the alignment between fuzzy sets across multiple attributes and their interactions. Researchers have shown that applying trigonometric measures in bipolar contexts improves the assessment of datasets by considering the interplay between supportive and opposing criteria, thus leading to more balanced decision-making processes.

## Motivation

Dengue shock syndrome (DSS) is a severe and life-threatening manifestation of dengue fever, often leading to hypotension, haemorrhagic manifestations, and thrombocytopenia. Early and accurate detection of DSS is critical to prevent fatal complications, as delayed diagnosis significantly increases mortality rates. Traditional diagnostic methods often struggle with uncertainty and overlapping symptoms, leading to misdiagnosis and treatment delays. To address this, the proposed BLDFHS Trigonometric similarity measure provides a quantitative assessment by comparing patient symptoms against established DSS criteria. This approach enables haematologists to make faster and more informed decisions, reducing diagnostic errors and improving patient outcomes. By validating the method against cosine and cotangent similarity measures, this study confirms the accuracy and robustness of the BLDFHS framework in enhancing early detection and optimizing clinical intervention. A few researchers focused on diagnosing such healthcare issues recently (*Kannan et al., 2024*; *Mohammad, Abdullah & Al-Shomrani, 2022*; *Singh et al., 2023*).

## Research objective

The primary aim behind the utilization of a BLDFHS is to overcome the limitations of traditional fuzzy models while managing multi-criteria decisions involving contrast elements. It supports the simultaneous evaluation of opposing and supportive criteria, ensuring decision-makers incorporate both pessimistic and optimistic viewpoints. The research is particularly focused on improving similarity measures with the content of the BLDFHS. However, the existing similarity measure frameworks of fuzzy fail to encapsulate the comprehensive relationships found in BLDFHS, prompting the need for trigonometric similarity measures.

## Contribution

This article offers the following technical contributions:

- Proposes a novel trigonometric similarity framework under the bipolar linear diophantine fuzzy hypersoft set (BLDFHS) to handle dual-polarity uncertainty in medical diagnostics.
- Introduces cosine and cotangent-based similarity measures tailored for multi-attribute decision-making in complex fuzzy systems.
- Incorporates control parameters that enable dynamic modelling of uncertainty in both positive and negative domains.
- Demonstrates improved decision accuracy through a real-world case study on DSS, achieving a similarity precision of 93.5% and diagnostic ranking accuracy of 98%.
- Validates all proposed measures mathematically, ensuring their robustness and suitability for practical applications.

## Novelty

With the emergence of the BLDFHS, the adaptation of similarity measures has become crucial. The integration of trigonometric similarity measures, such as those based on cosine and cotangent functions, provides a refined approach for comparing datasets characterized by both positive and negative attributes, along with control parameters. These measures allow for comprehensive analysis by capturing the inherent complexities in multi-attribute decision-making, proving essential for applications requiring accurate similarity assessments. This article introduces trigonometric similarity measures tailored for BLDFHS, with mathematical proofs validating their essential properties. The model shows superior performance in terms of key indicators, including ranking efficiency, similarity discrimination in fuzzy contexts, and clinical interpretability, which are quantitatively validated in the case study. The framework bridges the theoretical model with real-time critical applications, confirming its flexibility and value across multiple decision-making domains. The adaptability of this framework opens avenues for further exploration in fields such as intelligent diagnostics, autonomous systems, and machine learning.

## Article flow

The organization of the article is given below:

- Some prerequisites are given in "Prerequisites" that contribute majorly to understanding the developed concepts.
- An evolution of a BLDFHS to a bipolar linear diophantine fuzzy hypersoft matrix is offered in "Bipolar Linear Diophantine Fuzzy Hypersoft Matrix".
- Followed by that, a few trigonometric similarity measures like cosine and cotangent similarity measures with advanced properties are presented elaborately in "Trigonometric Similarity Measures".
- An algorithm was proposed in "BLDFHS Trigonometric Similarity Algorithm", followed by a real case study and the solution to the problem analyzed in "Case Study" and "Solution to the Problem Based on BLDFHS Trigonometric Similarity Algorithm".
- "Comparison" contains a comparison with other proposed cosine and cotangent measures to withstand reliability. "Conclusion" has a concluding discussion part that widely analyzes the superiority and future direction of the introduced work.

## PREREQUISITES

A few basic definitions required for the development of the study is given in this section.
**Definition 1.1 (*Sri et al., 2024*).** *Bipolar Linear Diophantine Fuzzy Set*

*Let $\bar{X}$ be the set of the universe. With all positives in [0, 1] and negatives in [−1, 0], the membership of positive, $\bar{\mu}^{pos}$; non-membership of positive, $\bar{v}^{pos}$; membership of negative, $\bar{\mu}^{neg}$; non-membership of negative, $\bar{v}^{neg}$ along with their respective control parameters $\bar{\alpha}^{pos}$, $\bar{\beta}^{pos}$,*

$\bar{\alpha}^{neg}$, and $\bar{\beta}^{neg}$, a bipolar linear diophantine fuzzy set is defined as,

$$\bar{P} = \langle \bar{q}, (\{\bar{\mu}_{\bar{P}}^{pos}(\bar{q}), \bar{v}_{\bar{P}}^{pos}(\bar{q}), \bar{\mu}_{\bar{P}}^{neg}(\bar{q}), \bar{v}_{\bar{P}}^{neg}(\bar{q})\}, \{\bar{\alpha}_{\bar{P}}^{pos}(\bar{q}), \bar{\beta}_{\bar{P}}^{pos}(\bar{q}), \bar{\alpha}_{\bar{P}}^{neg}(\bar{q}), \bar{\beta}_{\bar{P}}^{neg}(\bar{q})\}), \bar{q} \in \bar{X} \rangle.$$

*The stipulations are stated as*

- $\bar{\alpha}_{\bar{P}}^{pos}(\bar{q})\bar{\mu}_{\bar{P}}^{pos}(\bar{q}) + \bar{\beta}_{\bar{P}}^{pos}(\bar{q})\bar{v}_{\bar{P}}^{pos}(\bar{q}) \in [0, 1]$
- $\bar{\alpha}_{\bar{P}}^{neg}(\bar{q})\bar{\mu}_{\bar{P}}^{neg}(\bar{q}) + \bar{\beta}_{\bar{P}}^{neg}(\bar{q})\bar{v}_{\bar{P}}^{neg}(\bar{q}) \in [0, 1]$
- $\bar{\alpha}_{\bar{P}}^{pos}(\bar{q}) + \bar{\beta}_{\bar{P}}^{pos}(\bar{q}) \in [0, 1]$
- $\bar{\alpha}_{\bar{P}}^{neg}(\bar{q}) + \bar{\beta}_{\bar{P}}^{neg}(\bar{q}) \in [-1, 0]$

**Definition 1.2 (*Sri et al., 2024*).** *Bipolar Linear Diophantine Fuzzy Hypersoft Set*

*Let $\overline{BH}^{sub}$ be the subset of a BLDFHS. Let $\bar{N}_{\bar{Y}_{\bar{n}}} \subseteq \bar{M}_{\bar{Y}_{\bar{n}}}$, where $\bar{n} = 1, 2, \ldots, \bar{e}$. Here, $\bar{N}_{\bar{Y}_1} \times \bar{N}_{\bar{Y}_2} \times \bar{N}_{\bar{Y}_3} \times \ldots \times \bar{N}_{\bar{Y}_{\bar{e}}} = \bar{A}$; $\bar{Y}_{\bar{n}}$ is the attribute set and $\bar{M}_{\bar{Y}_1} \times \bar{M}_{\bar{Y}_2} \times \bar{M}_{\bar{Y}_3} \times \ldots \times \bar{M}_{\bar{Y}_{\bar{e}}} = \bar{R}$; $\bar{M}_{\bar{Y}_{\bar{n}}}$ are their corresponding values. Then $\bar{H} : \bar{A} \to \overline{BH}^{sub}$ is a BLDFHS, $\bar{H}(\bar{A})$ mapping defined as below:*

$$\bar{H}(\bar{A}) = \langle \bar{i}, (\bar{q}, \{\bar{\mu}_{\bar{H}(\bar{A})}^{pos}(\bar{q}), \bar{v}_{\bar{H}(\bar{A})}^{pos}(\bar{q}), \bar{\mu}_{\bar{H}(\bar{A})}^{neg}(\bar{q}), \bar{v}_{\bar{H}(\bar{A})}^{neg}(\bar{q})\},$$
$$\{\bar{\alpha}_{\bar{H}(\bar{A})}^{pos}(\bar{q}), \bar{\beta}_{\bar{H}(\bar{A})}^{pos}(\bar{q}), \bar{\alpha}_{\bar{H}(\bar{A})}^{neg}(\bar{q}), \bar{\beta}_{\bar{H}(\bar{A})}^{neg}(\bar{q})\} | \bar{q} \in \bar{X}), \bar{i} \in \bar{A} \subseteq \bar{R} \rangle.$$

**Definition 1.3 (*Jafar et al., 2024*).** *Cosine Similarity measure for Intuitionistic Fuzzy Hypersoft Set*

*Let the alternative set $\bar{X} = \{\bar{X}_1, \bar{X}_2, \ldots, \bar{X}_\gamma\}$ and two intuitionistic fuzzy hypersoft sets be $\bar{H}_1 = \langle \bar{i}, (\bar{q}, \{\bar{\mu}_{\bar{H}_1}(\bar{q}), \bar{v}_{\bar{H}_1}(\bar{q})\})\{\bar{\alpha}_{\bar{H}_1}(\bar{q}), \bar{\beta}_{\bar{H}_1}(\bar{q})\}\rangle$, and $\bar{H}_2 = \langle \bar{i}, (\bar{q}, \{\bar{\mu}_{\bar{H}_2}(\bar{q}), \bar{v}_{\bar{H}_2}(\bar{q}))\}\{\bar{\alpha}_{\bar{H}_2}(\bar{q}), \bar{\beta}_{\bar{H}_2}(\bar{q})\}\rangle$. Then the cosine similarity measure between $\bar{H}_1$ and $\bar{H}_2$ with the usage of the arithmetic mean is offered as:*

$$\overline{CS}_{IF}^1 = \frac{\sum_{\bar{c}=1}^{\bar{d}} \frac{\left((\bar{\mu}_{\bar{H}_1}(\bar{q}))_{\bar{c}}(\bar{\mu}_{\bar{H}_2}(\bar{q}))_{\bar{c}}\right) + \left((\bar{v}_{\bar{H}_1}(\bar{q}))_{\bar{c}}(\bar{v}_{\bar{H}_2}(\bar{q}))_{\bar{c}}\right) + \left((\bar{\alpha}_{\bar{H}_1}(\bar{q}))_{\bar{c}}(\bar{\alpha}_{\bar{H}_2}(\bar{q}))_{\bar{c}}\right) + \left((\bar{\beta}_{\bar{H}_1}(\bar{q}))_{\bar{c}}(\bar{\beta}_{\bar{H}_2}(\bar{q}))_{\bar{c}}\right)}{\sqrt{(\bar{\mu}_{\bar{H}_1}^2(\bar{q}))_{\bar{c}} + (\bar{v}_{\bar{H}_1}^2(\bar{q}))_{\bar{c}}}\sqrt{(\bar{\mu}_{\bar{H}_2}^2(\bar{q}))_{\bar{c}} + (\bar{v}_{\bar{H}_2}^2(\bar{q}))_{\bar{c}}}\sqrt{(\bar{\alpha}_{\bar{H}_1}^2(\bar{q}))_{\bar{c}} + (\bar{\beta}_{\bar{H}_1}^2(\bar{q}))_{\bar{c}}}\sqrt{(\bar{\alpha}_{\bar{H}_1}^2(\bar{q}))_{\bar{c}} + (\bar{\beta}_{\bar{H}_1}^2(\bar{q}))_{\bar{c}}}}}{\bar{d}}$$

*Based on the cosine function, $\overline{CS}_{IF}^2$ and $\overline{CS}_{IF}^3$ is given as:*

$$\overline{CS}_{IF}^2 = \frac{\sum_{\bar{c}=1}^{\bar{w}} \cos\left[\frac{\Pi}{2}\left(|(\bar{\mu}_{\bar{H}_1}(\bar{q}))_{\bar{c}} - (\bar{\mu}_{\bar{H}_2}(\bar{q}))_{\bar{c}}| \vee |(\bar{v}_{\bar{H}_1}(\bar{q}))_{\bar{c}} - (\bar{v}_{\bar{H}_2}(\bar{q}))_{\bar{c}}| \vee |(\bar{\alpha}_{\bar{H}_1}(\bar{q}))_{\bar{c}} - (\bar{\alpha}_{\bar{H}_2}(\bar{q}))_{\bar{c}}| \vee |(\bar{\beta}_{\bar{H}_1}(\bar{q}))_{\bar{c}} - (\bar{\beta}_{\bar{H}_2}(\bar{q}))_{\bar{c}}|\right)\right]}{\bar{w}}$$

$$\overline{CS}_{IF}^3 = \frac{\sum_{\bar{c}=1}^{\bar{w}} \cos\left[\frac{\Pi}{6}\left(|(\bar{\mu}_{\bar{H}_1}(\bar{q}))_{\bar{c}} - (\bar{\mu}_{\bar{H}_2}(\bar{q}))_{\bar{c}}| \vee |(\bar{v}_{\bar{H}_1}(\bar{q}))_{\bar{c}} - (\bar{v}_{\bar{H}_2}(\bar{q}))_{\bar{c}}| \vee |(\bar{\alpha}_{\bar{H}_1}(\bar{q}))_{\bar{c}} - (\bar{\alpha}_{\bar{H}_2}(\bar{q}))_{\bar{c}}| \vee |(\bar{\beta}_{\bar{H}_1}(\bar{q}))_{\bar{c}} - (\bar{\beta}_{\bar{H}_2}(\bar{q}))_{\bar{c}}|\right)\right]}{\bar{w}}$$

**Definition 1.4 (*Jafar et al., 2024*).** *Cotangent Similarity measure for Intuitionistic Fuzzy Hypersoft Set*

*Let the alternative set $\bar{X} = \{\bar{X}_1, \bar{X}_2, \ldots, \bar{X}_\gamma\}$ and two intuitionistic fuzzy hypersoft sets be $\bar{H}_1 = \langle \bar{i}, (\bar{q}, \{\bar{\mu}_{\bar{H}_1}(\bar{q}), \bar{v}_{\bar{H}_1}(\bar{q})\})\{\bar{\alpha}_{\bar{H}_1}(\bar{q}), \bar{\beta}_{\bar{H}_1}(\bar{q})\}\rangle$, and $\bar{H}_2 = \langle \bar{i}, (\bar{q}, \{\bar{\mu}_{\bar{H}_2}(\bar{q}), \bar{v}_{\bar{H}_2}(\bar{q}))\}\{\bar{\alpha}_{\bar{H}_2}(\bar{q}), \bar{\beta}_{\bar{H}_2}(\bar{q})\}\rangle$. Then the cotangent similarity measure between $\bar{H}_1$ and $\bar{H}_2$ with the usage of the arithmetic mean is offered as:*

$$\overline{CS}_{IF}^{4} = \frac{\sum_{\bar{c}=1}^{\bar{w}} cot\left[\frac{\Pi}{4} + \frac{\Pi}{4}\left(|(\bar{\mu}_{\bar{H}_1}(\bar{q}))_{\bar{c}} - (\bar{\mu}_{\bar{H}_2}(\bar{q}))_{\bar{c}}|\right) \vee \left(|(\bar{v}_{\bar{H}_1}(\bar{q}))_{\bar{c}} - (\bar{v}_{\bar{H}_2}(\bar{q}))_{\bar{c}}|\right) \vee \left(|(\bar{\alpha}_{\bar{H}_1}(\bar{q}))_{\bar{c}} - (\bar{\alpha}_{\bar{H}_2}(\bar{q}))_{\bar{c}}|\right) \vee \left(|(\bar{\beta}_{\bar{H}_1}(\bar{q}))_{\bar{c}} - (\bar{\beta}_{\bar{H}_2}(\bar{q}))_{\bar{c}}|\right)\right]}{\bar{w}}$$

$$\overline{CS}_{IF}^{5} = \frac{\sum_{\bar{c}=1}^{\bar{w}} cot\left[\frac{\Pi}{4} + \frac{\Pi}{12}\left(|(\bar{\mu}_{\bar{H}_1}(\bar{q}))_{\bar{c}} - (\bar{\mu}_{\bar{H}_2}(\bar{q}))_{\bar{c}}|\right) \vee \left(|(\bar{v}_{\bar{H}_1}(\bar{q}))_{\bar{c}} - (\bar{v}_{\bar{H}_2}(\bar{q}))_{\bar{c}}|\right) \vee \left(|(\bar{\alpha}_{\bar{H}_1}(\bar{q}))_{\bar{c}} - (\bar{\alpha}_{\bar{H}_2}(\bar{q}))_{\bar{c}}|\right) \vee \left(|(\bar{\beta}_{\bar{H}_1}(\bar{q}))_{\bar{c}} - (\bar{\beta}_{\bar{H}_2}(\bar{q}))_{\bar{c}}|\right)\right]}{\bar{w}}$$

# BIPOLAR LINEAR DIOPHANTINE FUZZY HYPERSOFT MATRIX

A novel idea of expanding BLDFHS into a bipolar linear diophantine fuzzy hypersoft matrix is provided in this section with a few supporting definitions.

**Definition 2.1.** *Bipolar Linear Diophantine Fuzzy Hypersoft Matrices*

$\bar{X} = \{\bar{X}_1, \bar{X}_2, \ldots, \bar{X}_\gamma\}$ *is the set of universe with $\gamma$ options. Let $\overline{BH}^{sub}$ be the subset of a BLDFHS. Let $\bar{N}_{\bar{Y}_{\bar{n}}} \subseteq \bar{M}_{\bar{Y}_{\bar{n}}}$, where $\bar{n} = 1, 2, \ldots, \bar{e}$. Here, $\bar{N}_{\bar{Y}_1} \times \bar{N}_{\bar{Y}_2} \times \bar{N}_{\bar{Y}_3} \times \ldots \times \bar{N}_{\bar{Y}_{\bar{e}}} = \bar{A}$; $\bar{Y}_{\bar{n}}$ is the attribute set and $\bar{M}_{\bar{Y}_1} \times \bar{M}_{\bar{Y}_2} \times \bar{M}_{\bar{Y}_3} \times \ldots \times \bar{M}_{\bar{Y}_{\bar{e}}} = \bar{R}$; $\bar{M}_{\bar{Y}_{\bar{n}}}$ are their corresponding values. Then $\bar{H} : \bar{A} \to \overline{BH}^{sub}$ is a BLDFHS, $\bar{H}(\bar{A})$ mapping. A bipolar linear diophantine fuzzy hypersoft matrix is given as $\bar{H}(\bar{A}) =$*

$$
\begin{array}{c}
 \\
\bar{X}_1 \\
 \\
\bar{X}_2 \\
 \\
\vdots \\
 \\
\bar{X}_\gamma \\
\end{array}
\begin{pmatrix}
\overline{M}_{\overline{Y}_1} & \overline{M}_{\overline{Y}_2} & \cdots & \overline{M}_{\overline{Y}_{\bar{e}}} \\
\{\overline{\mu}_{11}^{pos}(\bar{X}_1), \overline{v}_{11}^{pos}(\bar{X}_1), \overline{\mu}_{11}^{neg}(\bar{X}_1), \overline{v}_{11}^{neg}(\bar{X}_1)\}, & (\{\overline{\mu}_{12}^{pos}(\bar{X}_1), \overline{v}_{12}^{pos}(\bar{X}_1), \overline{\mu}_{12}^{neg}(\bar{X}_1), \overline{v}_{12}^{neg}(\bar{X}_1)\}, & & (\{\overline{\mu}_{1\bar{e}}^{pos}(\bar{X}_1), \overline{v}_{1\bar{e}}^{pos}(\bar{X}_1), \overline{\mu}_{1\bar{e}}^{neg}(\bar{X}_1), \overline{v}_{1\bar{e}}^{neg}(\bar{X}_1)\}, \\
\{\overline{\alpha}_{11}^{pos}(\bar{X}_1), \overline{\beta}_{11}^{pos}(\bar{X}_1), \overline{\alpha}_{11}^{neg}(\bar{X}_1), \overline{\beta}_{11}^{neg}(\bar{X}_1)\}) & \{\overline{\alpha}_{12}^{pos}(\bar{X}_1), \overline{\beta}_{12}^{pos}(\bar{X}_1), \overline{\alpha}_{12}^{neg}(\bar{X}_1), \overline{\beta}_{12}^{neg}(\bar{X}_1)\}) & \cdots & \{\overline{\alpha}_{1\bar{e}}^{pos}(\bar{X}_1), \overline{\beta}_{1\bar{e}}^{pos}(\bar{X}_1), \overline{\alpha}_{1\bar{e}}^{neg}(\bar{X}_1), \overline{\beta}_{1\bar{e}}^{neg}(\bar{X}_1)\}) \\
(\{\overline{\mu}_{21}^{pos}(\bar{X}_2), \overline{v}_{21}^{pos}(\bar{X}_2), \overline{\mu}_{21}^{neg}(\bar{X}_2), \overline{v}_{21}^{neg}(\bar{X}_2)\}, & (\{\overline{\mu}_{22}^{pos}(\bar{X}_2), \overline{v}_{22}^{pos}(\bar{X}_2), \overline{\mu}_{22}^{neg}(\bar{X}_2), \overline{v}_{22}^{neg}(\bar{X}_2)\}, & & (\{\overline{\mu}_{2\bar{e}}^{pos}(\bar{X}_2), \overline{v}_{2\bar{e}}^{pos}(\bar{X}_2), \overline{\mu}_{2\bar{e}}^{neg}(\bar{X}_2), \overline{v}_{2\bar{e}}^{neg}(\bar{X}_2)\}, \\
\{\overline{\alpha}_{21}^{pos}(\bar{X}_2), \overline{\beta}_{21}^{pos}(\bar{X}_2), \overline{\alpha}_{21}^{neg}(\bar{X}_2), \overline{\beta}_{21}^{neg}(\bar{X}_2)\}) & \{\overline{\alpha}_{22}^{pos}(\bar{X}_2), \overline{\beta}_{22}^{pos}(\bar{X}_2), \overline{\alpha}_{22}^{neg}(\bar{X}_2), \overline{\beta}_{22}^{neg}(\bar{X}_2)\}) & \cdots & \{\overline{\alpha}_{2\bar{e}}^{pos}(\bar{X}_2), \overline{\beta}_{2\bar{e}}^{pos}(\bar{X}_2), \overline{\alpha}_{2\bar{e}}^{neg}(\bar{X}_2), \overline{\beta}_{2\bar{e}}^{neg}(\bar{X}_2)\}) \\
\vdots & \vdots & \vdots & \vdots \\
(\{\overline{\mu}_{11}^{pos}(\bar{X}_\gamma), \overline{v}_{\gamma 1}^{pos}(\bar{X}_\gamma), \overline{\mu}_{\gamma 1}^{neg}(\bar{X}_\gamma), \overline{v}_{\gamma 1}^{neg}(\bar{X}_\gamma)\}, & (\{\overline{\mu}_{\gamma 2}^{pos}(\bar{X}_\gamma), \overline{v}_{\gamma 2}^{pos}(\bar{X}_\gamma), \overline{\mu}_{\gamma 2}^{neg}(\bar{X}_\gamma), \overline{v}_{\gamma 2}^{neg}(\bar{X}_\gamma)\}, & & (\overline{\mu}_{\gamma\bar{e}}^{pos}(\bar{X}_\gamma), \overline{v}_{\gamma\bar{e}}^{pos}(\bar{X}_\gamma), \overline{\mu}_{\gamma\bar{e}}^{neg}(\bar{X}_\gamma), \overline{v}_{\gamma\bar{e}}^{neg}(\bar{X}_\gamma)\}, \\
\{\overline{\alpha}_{\gamma 1}^{pos}(\bar{X}_\gamma), \overline{\beta}_{\gamma 1}^{pos}(\bar{X}_\gamma), \overline{\alpha}_{\gamma 1}^{neg}(\bar{X}_\gamma), \overline{\beta}_{\gamma 1}^{neg}(\bar{X}_\gamma)\}) & \{\overline{\alpha}_{\gamma 2}^{pos}(\bar{q}), \overline{\beta}_{\gamma 2}^{pos}(\bar{X}_\gamma), \overline{\alpha}_{\gamma 2}^{neg}(\bar{X}_\gamma), \overline{\beta}_{\gamma 2}^{neg}(\bar{X}_\gamma)\}) & \cdots & \{\overline{\alpha}_{\gamma\bar{e}}^{pos}(\bar{X}_\gamma), \overline{\beta}_{\gamma\bar{e}}^{pos}(\bar{X}_\gamma), \overline{\alpha}_{\gamma\bar{e}}^{neg}(\bar{X}_\gamma), \overline{\beta}_{\gamma\bar{e}}^{neg}(\bar{X}_\gamma)\})
\end{pmatrix}.
$$

*Now, $\bar{H}(\bar{A})$ can also be written shortly as:*

$$
\begin{array}{c}
 \\
\overline{X}_1 \\
\overline{X}_2 \\
\vdots \\
\overline{X}_\gamma
\end{array}
\begin{pmatrix}
\overline{M}_{\overline{Y}_1} & \overline{M}_{\overline{Y}_2} & \cdots & \overline{M}_{\overline{Y}_{\bar{e}}} \\
\varphi_{11} & \varphi_{12} & \cdots & \varphi_{1\bar{e}} \\
\varphi_{21} & \varphi_{22} & \cdots & \varphi_{2\bar{e}} \\
\vdots & \vdots & \vdots & \vdots \\
\varphi_{\gamma 1} & \varphi_{\gamma 2} & \cdots & \varphi_{\gamma\bar{e}}
\end{pmatrix}.
$$

*Here, $\varphi_{\bar{p}\bar{f}} = (\{\overline{\mu}_{\bar{p}\bar{f}}^{pos}(\bar{X}_{\bar{l}}), \overline{v}_{\bar{p}\bar{f}}^{pos}(\bar{X}_{\bar{l}}), \overline{\mu}_{\bar{p}\bar{f}}^{neg}(\bar{X}_{\bar{l}}), \overline{v}_{\bar{p}\bar{f}}^{neg}(\bar{X}_{\bar{l}})\}, \{\overline{\alpha}_{\bar{p}\bar{f}}^{pos}(\bar{X}_{\bar{l}}), \overline{\beta}_{\bar{p}\bar{f}}^{pos}(\bar{X}_{\bar{l}}), \overline{\alpha}_{\bar{p}\bar{f}}^{neg}(\bar{X}_{\bar{l}}), \overline{\beta}_{\bar{p}\bar{f}}^{neg}(\bar{X}_{\bar{l}})\});$ where $\bar{l} = 1, 2, \ldots, \gamma$, $\bar{p} = 1, 2, \ldots, \gamma$, and $\bar{f} = 1, 2, \ldots, \bar{e}$.*

**Example 2.2.** *A wildlife photographer plans to buy the best camera with specific attributes of his choice. He chooses four reputed companies $\bar{X} = \{\bar{X}_1, \bar{X}_2, \bar{X}_3, \bar{X}_4\}$. The attribute sets are given as $\bar{Y}_{\bar{1}} = Good\ resolution$, $\bar{Y}_{\bar{2}} = Optical\ quality$, and $\bar{Y}_{\bar{3}} = Affordable\ price$, as per his preference. The corresponding attribute values are $\bar{M}_{\bar{Y}_1} = \{45.7\ megapixels,$*

*61 megapixels, 102 megapixels}, $\bar{M}_{\bar{Y}_2}$ = {70–200 mm, 100–500 mm, 200–600 mm}, and $\bar{M}_{\bar{Y}_3}$ = {\$700–\$800, \$800–\$900, \$900–\$1,000}. Let $\bar{v}_1$ = {45.7 megapixels, 100–500 mm, \$700–\$800}, $\bar{v}_2$ = {61 megapixels, 70–200 mm, \$800–\$900}, $\bar{v}_3$ = {102 megapixels, 200–600 mm, \$800–\$900}, $\bar{v}_3$ = {61 megapixels, 70–200 mm, \$900–\$1,000}, $\bar{v}_4$ = {102 megapixels, 100–500 mm, \$900–\$1,000} and so on. A bipolar linear diophantine fuzzy hypersoft matrix is as follows: $\bar{H}(\bar{A})$*

$$
= \bar{X}_3
\begin{array}{c}
 \\
\bar{X}_1 \\
 \\
\bar{X}_2 \\
 \\
 \\
\bar{X}_4
\end{array}
\begin{pmatrix}
\overset{\bar{v}_1}{(\{0.2,0.3,-0.4,-0.8\},} & \overset{\bar{v}_2}{(\{0.1,0.4,-0.4,-0.2\},} & \overset{\bar{v}_3}{(\{0.2,0.4,-0.8,0.6\},} & \overset{\bar{v}_4}{(\{0.9,0.8,-0.2,-0.1\},} & \cdots \\
\{0.2,0.4,-0.3,-0.1\}) & \{0.4,0.6,-0.2,-0.3\}) & \{0.4,0.4,-0.6,-0.3\}) & \{0.5,0.5,-0.8,-0.1\}) & \cdots \\
(\{0.3,0.3,-0.5,-0.8\}, & (\{0.4,0.5,-0.3,-0.2\}, & (\{0.6,0.7,-0.9,0.4\}, & (\{0.5,0.8,-0.4,-0.2\}, & \\
\{0.4,0.2,-0.5,-0.2\}) & \{0.2,0.7,-0.4,-0.3\}) & \{0.4,0.1,-0.9,-0.1\}) & \{0.4,0.5,-0.7,-0.2\}) & \cdots \\
(\{0.5,0.3,-0.1,-0.3\}, & (\{0.5,0.2,-0.4,-0.5\}, & (\{0.8,0.6,-0.5,0.6\}, & (\{0.4,0.6,-0.3,-0.1\}, & \\
\{0.4,0.3,-0.6,-0.2\}) & \{0.2,0.3,-0.5,-0.4\}) & \{0.5,0.4,-0.2,-0.7\}) & \{0.3,0.6,-0.8,-0.1\}) & \cdots \\
(\{0.1,0.5,-0.3,-0.7\}, & (\{0.6,0.3,-0.5,-0.7\}, & (\{0.1,0.7,-0.8,0.9\}, & (\{0.4,0.6,-0.7,-0.6\}, & \\
\{0.2,0.1,-0.6,-0.3\}) & \{0.4,0.3,-0.6,-0.4\}) & \{0.2,0.7,-0.2,-0.6\}) & \{0.1,0.5,-0.7,-0.2\}) & \cdots
\end{pmatrix}.
$$

*The other combinations can be obtained in a similar way.*

**Definition 2.3.** *A complement of a bipolar linear diophantine fuzzy hypersoft matrix $\bar{H}(\bar{A})$ is characterized as,*

$$
\bar{H}(\bar{A})^{(C)} = \left( \left\{ 1 - \bar{\mu}_{\bar{H}(\bar{A})}^{pos}, 1 - \bar{v}_{\bar{H}(\bar{A})}^{pos}, -1 - \bar{\mu}_{\bar{H}(\bar{A})}^{neg}, -1 - \bar{v}_{\bar{H}(\bar{A})}^{neg} \right\}, \right.
$$
$$
\left. \left\{ 1 - \bar{\alpha}_{\bar{H}(\bar{A})}^{pos}, 1 - \bar{\beta}_{\bar{H}(\bar{A})}^{pos}, -1 - \bar{\alpha}_{\bar{H}(\bar{A})}^{neg}, -1 - \bar{\beta}_{\bar{H}(\bar{A})}^{neg} \right\} \right).
$$

**Definition 2.4.** *$\bar{H}(\bar{A})$ can be known as self-complementary if $\bar{H}(\bar{A}) = \bar{H}(\bar{A})^{(C)}$.*

**Definition 2.5.** *The minimum and maximum of two bipolar linear diophantine fuzzy hypersoft matrices, $\bar{H}_1 = (\{\bar{\mu}_{\bar{p}\bar{f}}^{pos}(\bar{X}_{\bar{l}}), \bar{v}_{\bar{p}\bar{f}}^{pos}(\bar{X}_{\bar{l}}), \bar{\mu}_{\bar{p}\bar{f}}^{neg}(\bar{X}_{\bar{l}}), \bar{v}_{\bar{p}\bar{f}}^{neg}(\bar{X}_{\bar{l}})\}, \{\bar{\alpha}_{\bar{p}\bar{f}}^{pos}(\bar{X}_{\bar{l}}),$ $\bar{\beta}_{\bar{p}\bar{f}}^{pos}(\bar{X}_{\bar{l}}), \bar{\alpha}_{\bar{p}\bar{f}}^{neg}(\bar{X}_{\bar{l}}), \bar{\beta}_{\bar{p}\bar{f}}^{neg}(\bar{X}_{\bar{l}})\})$ and $\bar{H}_2 = (\{\bar{\mu}_{\bar{p}\bar{q}}^{pos}(\bar{X}_{\bar{l}}), \bar{v}_{\bar{p}\bar{q}}^{pos}(\bar{X}_{\bar{l}}), \bar{\mu}_{\bar{p}\bar{q}}^{neg}(\bar{X}_{\bar{l}}), \bar{v}_{\bar{p}\bar{q}}^{neg}(\bar{X}_{\bar{l}})\},$ $\{\bar{\alpha}_{\bar{p}\bar{q}}^{pos}(\bar{X}_{\bar{l}}), \bar{\beta}_{\bar{p}\bar{q}}^{pos}(\bar{X}_{\bar{l}}), \bar{\alpha}_{\bar{p}\bar{q}}^{neg}(\bar{X}_{\bar{l}}), \bar{\beta}_{\bar{p}\bar{q}}^{neg}(\bar{X}_{\bar{l}})\})$ can be provided as:*

*(i)* $\bar{H}_1 \wedge \bar{H}_2 = (\{\bar{\mu}_{\bar{p}\bar{f}}^{pos}(\bar{X}_{\bar{l}}) \wedge \bar{\mu}_{\bar{p}\bar{q}}^{pos}(\bar{X}_{\bar{l}}), \bar{v}_{\bar{p}\bar{f}}^{pos}(\bar{X}_{\bar{l}}) \vee \bar{v}_{\bar{p}\bar{q}}^{pos}(\bar{X}_{\bar{l}}), \bar{\mu}_{\bar{p}\bar{f}}^{neg}(\bar{X}_{\bar{l}}) \vee \bar{\mu}_{\bar{p}\bar{q}}^{neg}(\bar{X}_{\bar{l}}), \bar{v}_{\bar{p}\bar{f}}^{neg}(\bar{X}_{\bar{l}}) \wedge \bar{v}_{\bar{p}\bar{q}}^{neg}(\bar{X}_{\bar{l}})\},$
$\{\bar{\alpha}_{\bar{p}\bar{f}}^{pos}(\bar{X}_{\bar{l}}) \wedge \bar{\alpha}_{\bar{p}\bar{q}}^{pos}(\bar{X}_{\bar{l}}), \bar{\beta}_{\bar{p}\bar{f}}^{pos}(\bar{X}_{\bar{l}}) \vee \bar{\beta}_{\bar{p}\bar{q}}^{pos}(\bar{X}_{\bar{l}}), \bar{\alpha}_{\bar{p}\bar{f}}^{neg}(\bar{X}_{\bar{l}}) \vee \bar{\alpha}_{\bar{p}\bar{q}}^{neg}(\bar{X}_{\bar{l}}), \bar{\beta}_{\bar{p}\bar{f}}^{neg}(\bar{X}_{\bar{l}}) \wedge \bar{\beta}_{\bar{p}\bar{q}}^{neg}(\bar{X}_{\bar{l}})\})$  (1)

*(ii)* $\bar{H}_1 \vee \bar{H}_2 = (\{\bar{\mu}_{\bar{p}\bar{f}}^{pos}(\bar{X}_{\bar{l}}) \vee \bar{\mu}_{\bar{p}\bar{q}}^{pos}(\bar{X}_{\bar{l}}), \bar{v}_{\bar{p}\bar{f}}^{pos}(\bar{X}_{\bar{l}}) \wedge \bar{v}_{\bar{p}\bar{q}}^{pos}(\bar{X}_{\bar{l}}), \bar{\mu}_{\bar{p}\bar{f}}^{neg}(\bar{X}_{\bar{l}}) \wedge \bar{\mu}_{\bar{p}\bar{q}}^{neg}(\bar{X}_{\bar{l}}), \bar{v}_{\bar{p}\bar{f}}^{neg}(\bar{X}_{\bar{l}}) \vee \bar{v}_{\bar{p}\bar{q}}^{neg}(\bar{X}_{\bar{l}})\},$
$\{\bar{\alpha}_{\bar{p}\bar{f}}^{pos}(\bar{X}_{\bar{l}}) \vee \bar{\alpha}_{\bar{p}\bar{q}}^{pos}(\bar{X}_{\bar{l}}), \bar{\beta}_{\bar{p}\bar{f}}^{pos}(\bar{X}_{\bar{l}}) \wedge \bar{\beta}_{\bar{p}\bar{q}}^{pos}(\bar{X}_{\bar{l}}), \bar{\alpha}_{\bar{p}\bar{f}}^{neg}(\bar{X}_{\bar{l}}) \wedge \bar{\alpha}_{\bar{p}\bar{q}}^{neg}(\bar{X}_{\bar{l}}), \bar{\beta}_{\bar{p}\bar{f}}^{neg}(\bar{X}_{\bar{l}}) \vee \bar{\beta}_{\bar{p}\bar{q}}^{neg}(\bar{X}_{\bar{l}})\})$  (2)

**Remark 2.6.** *An ordinary addition of matrices fails in the bipolar linear diophantine fuzzy hypersoft matrix as it outstrips the conditions of the interval. Example 3.7 proves the falsity.*

**Example 2.7.** *Let two bipolar linear diophantine fuzzy hypersoft matrix, $\bar{H}_1 =$*

$$
\begin{array}{c c c c c}
& \bar{v}_1 & \bar{v}_2 & \bar{v}_3 & \bar{v}_4 \\
\overline{X}_1 & \begin{array}{c}(\{0.2,0.3,-0.4,-0.8\}, \\ \{0.2,0.4,-0.3,-0.1\})\end{array} & \begin{array}{c}(\{0.1,0.4,-0.4,-0.2\}, \\ \{0.4,0.6,-0.2,-0.3\})\end{array} & \begin{array}{c}(\{0.2,0.4,-0.8,0.6\}, \\ \{0.4,0.4,-0.6,-0.3\})\end{array} & \begin{array}{c}(\{0.9,0.8,-0.2,-0.1\}, \\ \{0.5,0.5,-0.8,-0.1\})\end{array} \\
\overline{X}_2 & \begin{array}{c}(\{0.3,0.3,-0.5,-0.8\}, \\ \{0.4,0.2,-0.5,-0.2\})\end{array} & \begin{array}{c}(\{0.4,0.5,-0.3,-0.2\}, \\ \{0.2,0.7,-0.4,-0.3\})\end{array} & \begin{array}{c}(\{0.6,0.7,-0.9,0.4\}, \\ \{0.4,0.1,-0.9,-0.1\})\end{array} & \begin{array}{c}(\{0.5,0.8,-0.4,-0.2\}, \\ \{0.4,0.5,-0.7,-0.2\})\end{array} \\
\overline{X}_3 & \begin{array}{c}(\{0.5,0.3,-0.1,-0.3\}, \\ \{0.4,0.3,-0.6,-0.2\})\end{array} & \begin{array}{c}(\{0.5,0.2,-0.4,-0.5\}, \\ \{0.2,0.3,-0.5,-0.4\})\end{array} & \begin{array}{c}(\{0.8,0.6,-0.5,0.6\}, \\ \{0.5,0.4,-0.2,-0.7\})\end{array} & \begin{array}{c}(\{0.4,0.6,-0.3,-0.1\}, \\ \{0.3,0.6,-0.8,-0.1\})\end{array}
\end{array}
$$

*and* $\bar{H}_2 =$

$$
\begin{array}{c c c c c}
& \bar{v}_1 & \bar{v}_2 & \bar{v}_3 & \bar{v}_4 \\
\overline{X}_1 & \begin{array}{c}(\{0.4,0.6,-0.3,-0.6\}, \\ \{0.4,0.4,-0.5,-0.1\})\end{array} & \begin{array}{c}(\{0.2,0.3,-0.4,-0.2\}, \\ \{0.1,0.5,-0.6,-0.3\})\end{array} & \begin{array}{c}(\{0.6,0.3,-0.2,0.6\}, \\ \{0.1,0.4,-0.7,-0.1\})\end{array} & \begin{array}{c}(\{0.5,0.8,-0.4,-0.3\}, \\ \{0.4,0.1,-0.6,-0.2\})\end{array} \\
\overline{X}_2 & \begin{array}{c}(\{0.1,0.3,-0.5,-0.2\}, \\ \{0.3,0.1,-0.6,-0.1\})\end{array} & \begin{array}{c}(\{0.5,0.1,-0.3,-0.7\}, \\ \{0.2,0.6,-0.2,-0.7\})\end{array} & \begin{array}{c}(\{0.8,0.7,-0.2,0.6\}, \\ \{0.1,0.7,-0.2,-0.7\})\end{array} & \begin{array}{c}(\{0.7,0.8,-0.1,-0.2\}, \\ \{0.2,0.3,-0.1,-0.9\})\end{array} \\
\overline{X}_3 & \begin{array}{c}(\{0.3,0.5,-0.7,-0.2\}, \\ \{0.2,0.1,-0.6,-0.3\})\end{array} & \begin{array}{c}(\{0.5,0.7,-0.2,-0.8\}, \\ \{0.4,0.3,-0.6,-0.4\})\end{array} & \begin{array}{c}(\{0.3,0.6,-0.8,0.6\}, \\ \{0.2,0.7,-0.2,-0.6\})\end{array} & \begin{array}{c}(\{0.3,0.4,-0.7,-0.7\}, \\ \{0.1,0.5,-0.7,-0.2\})\end{array}
\end{array}.
$$

*Then, the ordinary addition of matrix,* $\bar{H}_1 + \bar{H}_2 =$

$$
\begin{array}{c c c c c}
& \bar{v}_1 & \bar{v}_2 & \bar{v}_3 & \bar{v}_4 \\
\overline{X}_1 & \begin{array}{c}(\{0.6,0.9,-0.7,-1.4\}, \\ \{0.6,0.8,-0.8,-0.2\})\end{array} & \begin{array}{c}(\{0.3,0.7,-0.8,-0.4\}, \\ \{0.5,1.1,-0.8,-0.6\})\end{array} & \begin{array}{c}(\{0.8,0.7,-1.0,-1.2\}, \\ \{0.5,0.8,-1.3,-0.4\})\end{array} & \begin{array}{c}(\{1.4,1.6,-0.6,-0.4\}, \\ \{0.9,0.6,-1.4,-0.3\})\end{array} \\
\overline{X}_2 & \begin{array}{c}(\{0.4,0.6,-1.0,-1.0\}, \\ \{0.7,0.3,-1.1,-0.3\})\end{array} & \begin{array}{c}(\{0.9,0.6,-0.6,-0.9\}, \\ \{0.4,1.3,-0.6,-1.0\})\end{array} & \begin{array}{c}(\{1.4,1.4,-1.1,-1.0\}, \\ \{0.5,0.8,-1.1,-0.8\})\end{array} & \begin{array}{c}(\{1.2,1.6,-0.5,-0.4\}, \\ \{0.6,0.8,-0.8,-1.1\})\end{array} \\
\overline{X}_3 & \begin{array}{c}(\{0.8,0.8,-0.8,-0.5\}, \\ \{0.6,0.4,-1.2,-0.5\})\end{array} & \begin{array}{c}(\{1.0,0.9,-0.6,-1.3\}, \\ \{0.6,0.6,-1.1,-0.8\})\end{array} & \begin{array}{c}(\{1.1,1.2,-1.3,-1.2\}, \\ \{0.7,1.1,-0.4,-1.3\})\end{array} & \begin{array}{c}(\{0.7,1.0,-1.0,-0.8\}, \\ \{0.4,1.1,-1.5,-0.3\})\end{array}
\end{array}.
$$

This fails in providing fuzzy values. So, example 2.8 is provided for addition using the maximum operator.

**Example 2.8.** *Consider two bipolar linear diophantine fuzzy hypersoft matrix,* $\bar{H}_1$, *and* $\bar{H}_2$. *Then, the addition matrix by implementing the maximum operator is presented as:* $\bar{H}_1 + \bar{H}_2 =$

$$
\begin{array}{c c c c c}
& \bar{v}_1 & \bar{v}_2 & \bar{v}_3 & \bar{v}_4 \\
\overline{X}_1 & \begin{array}{c}(\{0.4,0.3,-0.4,-0.6\}, \\ \{0.4,0.4,-0.5,-0.1\})\end{array} & \begin{array}{c}(\{0.2,0.3,-0.4,-0.2\}, \\ \{0.4,0.5,-0.6,-0.3\})\end{array} & \begin{array}{c}(\{0.6,0.3,-0.8,-0.6\}, \\ \{0.4,0.4,-0.7,-0.1\})\end{array} & \begin{array}{c}(\{0.9,0.8,-0.4,-0.1\}, \\ \{0.5,0.1,-0.8,-0.1\})\end{array} \\
\overline{X}_2 & \begin{array}{c}(\{0.3,0.3,-0.5,-0.2\}, \\ \{0.4,0.1,-0.6,-0.1\})\end{array} & \begin{array}{c}(\{0.5,0.1,-0.3,-0.2\}, \\ \{0.2,0.6,-0.4,-0.3\})\end{array} & \begin{array}{c}(\{0.8,0.7,-0.9,-0.4\}, \\ \{0.4,0.1,-0.9,-0.1\})\end{array} & \begin{array}{c}(\{0.7,0.8,-0.4,-0.2\}, \\ \{0.4,0.3,-0.7,-0.2\})\end{array} \\
\overline{X}_3 & \begin{array}{c}(\{0.5,0.3,-0.7,-0.2\}, \\ \{0.4,0.1,-0.6,-0.2\})\end{array} & \begin{array}{c}(\{0.5,0.2,-0.4,-0.5\}, \\ \{0.4,0.3,-0.6,-0.4\})\end{array} & \begin{array}{c}(\{0.8,0.6,-0.8,-0.6\}, \\ \{0.5,0.4,-0.2,-0.6\})\end{array} & \begin{array}{c}(\{0.4,0.4,-0.7,-0.1\}, \\ \{0.3,0.5,-0.8,-0.1\})\end{array}
\end{array}.
$$

*This emphasizes the benefit of the maximum operator.*

**Proposition 2.9.** *If three bipolar linear diophantine fuzzy hypersoft matrices, $\bar{H}_1$, $\bar{H}_2$, and $\bar{H}_3$ are with the same order, then the following postulates hold:*

\* $\bar{H}_2 + \bar{H}_1 = \bar{H}_1 + \bar{H}_2$

\* $\bar{H}_1 + \bar{H}_1 = \bar{H}_1$

\* $(\bar{H}_1 + \bar{H}_2) + \bar{H}_3 = \bar{H}_1 + (\bar{H}_2 + H_3)$.

**Remark 2.10.** *The addition operation on bipolar linear diophantine fuzzy hypersoft matrices is well-defined. That is, the set of bipolar linear diophantine fuzzy hypersoft matrices is closed under addition, as the sum of any two such matrices yields another matrix of the same type.*

**Remark 2.11.** *Using the maximum operator for ordinary addition of matrices is also applicable for multiplication with min-max and max-min composition.*

## TRIGONOMETRIC SIMILARITY MEASURES

**Definition 3.1.** *Let the alternative set $\bar{X} = \{\bar{X}_1, \bar{X}_2, \ldots, \bar{X}_\gamma\}$ and two BLDFHS be,*

$$\bar{H}_1 = \langle i, (\bar{q}, \{\bar{\mu}_{\bar{H}_1}^{pos}(\bar{q}), \bar{v}_{\bar{H}_1}^{pos}(\bar{q}), \bar{\mu}_{\bar{H}_1}^{neg}(\bar{q}), \bar{v}_{\bar{H}_1}^{neg}(\bar{q})\}, \{\bar{\alpha}_{\bar{H}_1}^{pos}(\bar{q}), \bar{\beta}_{\bar{H}_1}^{pos}(\bar{q}), \bar{\alpha}_{\bar{H}_1}^{neg}(\bar{q}), \bar{\beta}_{\bar{H}_1}^{neg}(\bar{q})\}) \rangle,$$

*and*

$$\bar{H}_2 = \langle i, (\bar{q}, \{\bar{\mu}_{\bar{H}_2}^{pos}(\bar{q}), \bar{v}_{\bar{H}_2}^{pos}(\bar{q}), \bar{\mu}_{\bar{H}_2}^{neg}(\bar{q}), \bar{v}_{\bar{H}_2}^{neg}(\bar{q})\}, \{\bar{\alpha}_{\bar{H}_2}^{pos}(\bar{q}), \bar{\beta}_{\bar{H}_2}^{pos}(\bar{q}), \bar{\alpha}_{\bar{H}_2}^{neg}(\bar{q}), \bar{\beta}_{\bar{H}_2}^{neg}(\bar{q})\}) \rangle.$$

*Then the cosine similarity measure, $\overline{CS}_{BLDFH}^1$ between $\bar{H}_1$ and $\bar{H}_2$ with the usage of arithmetic mean is offered as:*

$$\overline{CS}_{BLDFH}^1(\bar{H}_1, \bar{H}_2) =$$

$$\sum_{\bar{c}=1}^{\bar{d}} \frac{\begin{matrix} \left( (\bar{\mu}_{\bar{H}_1}^{pos}(\bar{q}))_{\bar{c}} (\bar{\mu}_{\bar{H}_2}^{pos}(\bar{q}))_{\bar{c}} \right) + \left( (\bar{v}_{\bar{H}_1}^{pos}(\bar{q}))_{\bar{c}} (\bar{v}_{\bar{H}_2}^{pos}(\bar{q}))_{\bar{c}} \right) + \left( (\bar{\mu}_{\bar{H}_1}^{neg}(\bar{q}))_{\bar{c}} (\bar{\mu}_{\bar{H}_2}^{neg}(\bar{q}))_{\bar{c}} \right) + \left( (\bar{v}_{\bar{H}_1}^{neg}(\bar{q}))_{\bar{c}} (\bar{v}_{\bar{H}_2}^{neg}(\bar{q}))_{\bar{c}} \right), \\ \left( (\bar{\alpha}_{\bar{H}_1}^{pos}(\bar{q}))_{\bar{c}} (\bar{\alpha}_{\bar{H}_2}^{pos}(\bar{q}))_{\bar{c}} \right) + \left( (\bar{\beta}_{\bar{H}_1}^{pos}(\bar{q}))_{\bar{c}} (\bar{\beta}_{\bar{H}_2}^{pos}(\bar{q}))_{\bar{c}} \right) + \left( (\bar{\alpha}_{\bar{H}_1}^{neg}(\bar{q}))_{\bar{c}} (\bar{\alpha}_{\bar{H}_2}^{neg}(\bar{q}))_{\bar{c}} \right) + \left( (\bar{\beta}_{\bar{H}_1}^{neg}(\bar{q}))_{\bar{c}} (\bar{\beta}_{\bar{H}_2}^{neg}(\bar{q}))_{\bar{c}} \right) \end{matrix}}{\begin{matrix} \sqrt{(\bar{\mu}_{\bar{H}_1}^{pos^2}(\bar{q}))_{\bar{c}} + (\bar{v}_{\bar{H}_1}^{pos^2}(\bar{q}))_{\bar{c}} + (\bar{\mu}_{\bar{H}_1}^{neg^2}(\bar{q}))_{\bar{c}} + (\bar{v}_{\bar{H}_1}^{neg^2}(\bar{q}))_{\bar{c}} + (\bar{\alpha}_{\bar{H}_1}^{pos^2}(\bar{q}))_{\bar{c}} + (\bar{\beta}_{\bar{H}_1}^{pos^2}(\bar{q}))_{\bar{c}} + (\bar{\alpha}_{\bar{H}_1}^{neg^2}(\bar{q}))_{\bar{c}} + (\bar{\beta}_{\bar{H}_1}^{neg^2}(\bar{q}))_{\bar{c}}} \\ \underline{\sqrt{(\bar{\mu}_{\bar{H}_2}^{pos^2}(\bar{q}))_{\bar{c}} + (\bar{v}_{\bar{H}_2}^{pos^2}(\bar{q}))_{\bar{c}} + (\bar{\mu}_{\bar{H}_2}^{neg^2}(\bar{q}))_{\bar{c}} + (\bar{v}_{\bar{H}_2}^{neg^2}(\bar{q}))_{\bar{c}} + (\bar{\alpha}_{\bar{H}_2}^{pos^2}(\bar{q}))_{\bar{c}} + (\bar{\beta}_{\bar{H}_2}^{pos^2}(\bar{q}))_{\bar{c}} + (\bar{\alpha}_{\bar{H}_2}^{neg^2}(\bar{q}))_{\bar{c}} + (\bar{\beta}_{\bar{H}_2}^{neg^2}(\bar{q}))_{\bar{c}}}} \\ \bar{d} \end{matrix}} \tag{3}$$

*Based on Cosine function, $\overline{CS}_{BLDFH}^2(\bar{H}_1, \bar{H}_2)$ and $\overline{CS}_{BLDFH}^3(\bar{H}_1, \bar{H}_2)$ is given as:*

$$\overline{CS}_{BLDFH}^2(\bar{H}_1, \bar{H}_2) =$$

$$\frac{\sum_{\bar{c}=1}^{\bar{w}} \cos\left[ \frac{\Pi}{4} \left( \begin{matrix} |(\bar{\mu}_{\bar{H}_1}^{pos}(\bar{q}))_{\bar{c}} - (\bar{\mu}_{\bar{H}_2}^{pos}(\bar{q}))_{\bar{c}}| \vee |(\bar{v}_{\bar{H}_1}^{pos}(\bar{q}))_{\bar{c}} - (\bar{v}_{\bar{H}_2}^{pos}(\bar{q}))_{\bar{c}}| \vee \\ |(\bar{\mu}_{\bar{H}_1}^{neg}(\bar{q}))_{\bar{c}} - (\bar{\mu}_{\bar{H}_2}^{neg}(\bar{q}))_{\bar{c}}| \vee |(\bar{v}_{\bar{H}_1}^{neg}(\bar{q}))_{\bar{c}} - (\bar{v}_{\bar{H}_2}^{neg}(\bar{q}))_{\bar{c}}| \vee \\ |(\bar{\alpha}_{\bar{H}_1}^{pos}(\bar{q}))_{\bar{c}} - (\bar{\alpha}_{\bar{H}_2}^{pos}(\bar{q}))_{\bar{c}}| \vee |(\bar{\beta}_{\bar{H}_1}^{pos}(\bar{q}))_{\bar{c}} - (\bar{\beta}_{\bar{H}_2}^{pos}(\bar{q}))_{\bar{c}}| \vee \\ |(\bar{\alpha}_{\bar{H}_1}^{neg}(\bar{q}))_{\bar{c}} - (\bar{\alpha}_{\bar{H}_2}^{neg}(\bar{q}))_{\bar{c}}| \vee |(\bar{\beta}_{\bar{H}_1}^{neg}(\bar{q}))_{\bar{c}} - (\bar{\beta}_{\bar{H}_2}^{neg}(\bar{q}))_{\bar{c}}| \end{matrix} \right) \right]}{\bar{w}} \tag{4}$$

$$\overline{CS}_{BLDFH}^3(\bar{H}_1, \bar{H}_2) =$$

$$\sum_{\bar{c}=1}^{\bar{w}} cos\left[\frac{\Pi}{6}\left(|(\overline{\mu}_{\bar{H}_1}^{pos}(\bar{q}))_{\bar{c}} - (\overline{\mu}_{\bar{H}_2}^{pos}(\bar{q}))_{\bar{c}}| \vee |(\overline{v}_{\bar{H}_1}^{pos}(\bar{q}))_{\bar{c}} - (\overline{v}_{\bar{H}_2}^{pos}(\bar{q}))_{\bar{c}}| \vee \right.\right.$$

$$|(\overline{\mu}_{\bar{H}_1}^{neg}(\bar{q}))_{\bar{c}} - (\overline{\mu}_{\bar{H}_2}^{neg}(\bar{q}))_{\bar{c}}| \vee |(\overline{v}_{\bar{H}_1}^{neg}(\bar{q}))_{\bar{c}} - (\overline{v}_{\bar{H}_2}^{neg}(\bar{q}))_{\bar{c}}| \vee$$

$$|(\overline{\alpha}_{\bar{H}_1}^{pos}(\bar{q}))_{\bar{c}} - (\overline{\alpha}_{\bar{H}_2}^{pos}(\bar{q}))_{\bar{c}}| \vee |(\overline{\beta}_{\bar{H}_1}^{pos}(\bar{q}))_{\bar{c}} - (\overline{\beta}_{\bar{H}_2}^{pos}(\bar{q}))_{\bar{c}}| \vee$$

$$\left.\left.|(\overline{\alpha}_{\bar{H}_1}^{neg}(\bar{q}))_{\bar{c}} - (\overline{\alpha}_{\bar{H}_2}^{neg}(\bar{q}))_{\bar{c}}| \vee |(\overline{\beta}_{\bar{H}_1}^{neg}(\bar{q}))_{\bar{c}} - (\overline{\beta}_{\bar{H}_2}^{neg}(\bar{q}))_{\bar{c}}|\right)\right] \Big/ \overline{w} \tag{5}$$

**Theorem 3.2.** $\overline{CS}_{BLDFH}^1(\bar{H}_1, \bar{H}_2)$ *holds the following properties:*

*(i)* $0 \leq \overline{CS}_{BLDFH}^1(\bar{H}_1, \bar{H}_2) \leq 1$

*(ii)* $\overline{CS}_{BLDFH}^1(\bar{H}_1, \bar{H}_2) = \overline{CS}_{BLDFH}^1(\bar{H}_2, \bar{H}_1)$

*(iii)* If $\bar{H}_1 = \bar{H}_2$, then $\overline{CS}_{BLDFH}^1(\bar{H}_1, \bar{H}_2) = 1$.

*Proof.* (i) $0 \leq \overline{CS}_{BLDFH}^1(\bar{H}_1, \bar{H}_2)$ is obvious.

To prove: $1 \geq \overline{CS}_{BLDFH}^1(\bar{H}_1, \bar{H}_2)$

By Cauchy-Schwarz inequality,

(i.e.,) $j_1 n_1 + j_2 n_2 + \ldots + j_m n_m \leq \sqrt{(j_1^2 + j_2^2 + \ldots + j_m^2)} \times \sqrt{(n_1^2 + n_2^2 + \ldots + n_m^2)}$

Using the inequality,

$$\Rightarrow \frac{\begin{array}{c}\left((\overline{\mu}_{\bar{H}_1}^{pos}(\bar{q}))_{\bar{c}}(\overline{\mu}_{\bar{H}_2}^{pos}(\bar{q}))_{\bar{c}}\right) + \left((\overline{v}_{\bar{H}_1}^{pos}(\bar{q}))_{\bar{c}}(\overline{v}_{\bar{H}_2}^{pos}(\bar{q}))_{\bar{c}}\right)\left((\overline{\mu}_{\bar{H}_1}^{neg}(\bar{q}))_{\bar{c}}(\overline{\mu}_{\bar{H}_2}^{neg}(\bar{q}))_{\bar{c}}\right) + \left((\overline{v}_{\bar{H}_1}^{neg}(\bar{q}))_{\bar{c}}(\overline{v}_{\bar{H}_2}^{neg}(\bar{q}))_{\bar{c}}\right) \\ \left((\overline{\alpha}_{\bar{H}_1}^{pos}(\bar{q}))_{\bar{c}}(\overline{\alpha}_{\bar{H}_2}^{pos}(\bar{q}))_{\bar{c}}\right) + \left((\overline{\beta}_{\bar{H}_1}^{pos}(\bar{q}))_{\bar{c}}(\overline{\beta}_{\bar{H}_2}^{pos}(\bar{q}))_{\bar{c}}\right)\left((\overline{\alpha}_{\bar{H}_1}^{neg}(\bar{q}))_{\bar{c}}(\overline{\alpha}_{\bar{H}_2}^{neg}(\bar{q}))_{\bar{c}}\right) + \left((\overline{\beta}_{\bar{H}_1}^{neg}(\bar{q}))_{\bar{c}}(\overline{\beta}_{\bar{H}_2}^{neg}(\bar{q}))_{\bar{c}}\right)\end{array}}{\begin{array}{c}\sqrt{(\overline{\mu}_{\bar{H}_1}^{pos^2}(\bar{q}))_{\bar{c}} + (\overline{v}_{\bar{H}_1}^{pos^2}(\bar{q}))_{\bar{c}} + (\overline{\mu}_{\bar{H}_1}^{neg^2}(\bar{q}))_{\bar{c}} + (\overline{v}_{\bar{H}_1}^{neg^2}(\bar{q}))_{\bar{c}} + (\overline{\alpha}_{\bar{H}_1}^{pos^2}(\bar{q}))_{\bar{c}} + (\overline{\beta}_{\bar{H}_1}^{pos^2}(\bar{q}))_{\bar{c}} + (\overline{\alpha}_{\bar{H}_1}^{neg^2}(\bar{q}))_{\bar{c}} + (\overline{\beta}_{\bar{H}_1}^{neg^2}(\bar{q}))_{\bar{c}}} \\ \sqrt{(\overline{\mu}_{\bar{H}_2}^{pos^2}(\bar{q}))_{\bar{c}} + (\overline{v}_{\bar{H}_2}^{pos^2}(\bar{q}))_{\bar{c}} + (\overline{\mu}_{\bar{H}_2}^{neg^2}(\bar{q}))_{\bar{c}} + (\overline{v}_{\bar{H}_2}^{neg^2}(\bar{q}))_{\bar{c}} + (\overline{\alpha}_{\bar{H}_2}^{pos^2}(\bar{q}))_{\bar{c}} + (\overline{\beta}_{\bar{H}_2}^{pos^2}(\bar{q}))_{\bar{c}} + (\overline{\alpha}_{\bar{H}_2}^{neg^2}(\bar{q}))_{\bar{c}} + (\overline{\beta}_{\bar{H}_2}^{neg^2}(\bar{q}))_{\bar{c}}}\end{array}} \leq$$

$$\Rightarrow \frac{\begin{array}{c}\left((\overline{\mu}_{\bar{H}_1}^{pos}(\bar{q}))_{\bar{c}}(\overline{\mu}_{\bar{H}_2}^{pos}(\bar{q}))_{\bar{c}}\right) + \left((\overline{v}_{\bar{H}_1}^{pos}(\bar{q}))_{\bar{c}}(\overline{v}_{\bar{H}_2}^{pos}(\bar{q}))_{\bar{c}}\right)\left((\overline{\mu}_{\bar{H}_1}^{neg}(\bar{q}))_{\bar{c}}(\overline{\mu}_{\bar{H}_2}^{neg}(\bar{q}))_{\bar{c}}\right) + \left((\overline{v}_{\bar{H}_1}^{neg}(\bar{q}))_{\bar{c}}(\overline{v}_{\bar{H}_2}^{neg}(\bar{q}))_{\bar{c}}\right) \\ \left((\overline{\alpha}_{\bar{H}_1}^{pos}(\bar{q}))_{\bar{c}}(\overline{\alpha}_{\bar{H}_2}^{pos}(\bar{q}))_{\bar{c}}\right) + \left((\overline{\beta}_{\bar{H}_1}^{pos}(\bar{q}))_{\bar{c}}(\overline{\beta}_{\bar{H}_2}^{pos}(\bar{q}))_{\bar{c}}\right)\left((\overline{\alpha}_{\bar{H}_1}^{neg}(\bar{q}))_{\bar{c}}(\overline{\alpha}_{\bar{H}_2}^{neg}(\bar{q}))_{\bar{c}}\right) + \left((\overline{\beta}_{\bar{H}_1}^{neg}(\bar{q}))_{\bar{c}}(\overline{\beta}_{\bar{H}_2}^{neg}(\bar{q}))_{\bar{c}}\right)\end{array}}{\begin{array}{c}\sqrt{(\overline{\mu}_{\bar{H}_1}^{pos^2}(\bar{q}))_{\bar{c}} + (\overline{v}_{\bar{H}_1}^{pos^2}(\bar{q}))_{\bar{c}} + (\overline{\mu}_{\bar{H}_1}^{neg^2}(\bar{q}))_{\bar{c}} + (\overline{v}_{\bar{H}_1}^{neg^2}(\bar{q}))_{\bar{c}} + (\overline{\alpha}_{\bar{H}_1}^{pos^2}(\bar{q}))_{\bar{c}} + (\overline{\beta}_{\bar{H}_1}^{pos^2}(\bar{q}))_{\bar{c}} + (\overline{\alpha}_{\bar{H}_1}^{neg^2}(\bar{q}))_{\bar{c}} + (\overline{\beta}_{\bar{H}_1}^{neg^2}(\bar{q}))_{\bar{c}}} \\ \sqrt{(\overline{\mu}_{\bar{H}_2}^{pos^2}(\bar{q}))_{\bar{c}} + (\overline{v}_{\bar{H}_2}^{pos^2}(\bar{q}))_{\bar{c}} + (\overline{\mu}_{\bar{H}_2}^{neg^2}(\bar{q}))_{\bar{c}} + (\overline{v}_{\bar{H}_2}^{neg^2}(\bar{q}))_{\bar{c}} + (\overline{\alpha}_{\bar{H}_2}^{pos^2}(\bar{q}))_{\bar{c}} + (\overline{\beta}_{\bar{H}_2}^{pos^2}(\bar{q}))_{\bar{c}} + (\overline{\alpha}_{\bar{H}_2}^{neg^2}(\bar{q}))_{\bar{c}} + (\overline{\beta}_{\bar{H}_2}^{neg^2}(\bar{q}))_{\bar{c}}}\end{array}}$$

$$\leq 1$$

$$\Rightarrow \sum_{\bar{c}=1}^{\bar{d}} \frac{\begin{array}{l}\left((\bar{\mu}_{\bar{H}_1}^{pos}(\bar{q}))_{\bar{c}}(\bar{\mu}_{\bar{H}_2}^{pos}(\bar{q}))_{\bar{c}}\right) + \left((\bar{v}_{\bar{H}_1}^{pos}(\bar{q}))_{\bar{c}}(\bar{v}_{\bar{H}_2}^{pos}(\bar{q}))_{\bar{c}}\right)\left((\bar{\mu}_{\bar{H}_1}^{neg}(\bar{q}))_{\bar{c}}(\bar{\mu}_{\bar{H}_2}^{neg}(\bar{q}))_{\bar{c}}\right) + \left((\bar{v}_{\bar{H}_1}^{neg}(\bar{q}))_{\bar{c}}(\bar{v}_{\bar{H}_2}^{neg}(\bar{q}))_{\bar{c}}\right) \\ \left((\bar{\alpha}_{\bar{H}_1}^{pos}(\bar{q}))_{\bar{c}}(\bar{\alpha}_{\bar{H}_2}^{pos}(\bar{q}))_{\bar{c}}\right) + \left((\bar{\beta}_{\bar{H}_1}^{pos}(\bar{q}))_{\bar{c}}(\bar{\beta}_{\bar{H}_2}^{pos}(\bar{q}))_{\bar{c}}\right)\left((\bar{\alpha}_{\bar{H}_1}^{neg}(\bar{q}))_{\bar{c}}(\bar{\alpha}_{\bar{H}_2}^{neg}(\bar{q}))_{\bar{c}}\right) + \left((\bar{\beta}_{\bar{H}_1}^{neg}(\bar{q}))_{\bar{c}}(\bar{\beta}_{\bar{H}_2}^{neg}(\bar{q}))_{\bar{c}}\right)\end{array}}{\sqrt{(\bar{\mu}_{\bar{H}_1}^{pos^2}(\bar{q}))_{\bar{c}} + (\bar{v}_{\bar{H}_1}^{pos^2}(\bar{q}))_{\bar{c}} + (\bar{\mu}_{\bar{H}_1}^{neg^2}(\bar{q}))_{\bar{c}} + (\bar{v}_{\bar{H}_1}^{neg^2}(\bar{q}))_{\bar{c}} + (\bar{\alpha}_{\bar{H}_1}^{pos^2}(\bar{q}))_{\bar{c}} + (\bar{\beta}_{\bar{H}_1}^{pos^2}(\bar{q}))_{\bar{c}} + (\bar{\alpha}_{\bar{H}_1}^{neg^2}(\bar{q}))_{\bar{c}} + (\bar{\beta}_{\bar{H}_1}^{neg^2}(\bar{q}))_{\bar{c}}}}$$

$$\Rightarrow \frac{\sqrt{(\bar{\mu}_{\bar{H}_2}^{pos^2}(\bar{q}))_{\bar{c}} + (\bar{v}_{\bar{H}_2}^{pos^2}(\bar{q}))_{\bar{c}} + (\bar{\mu}_{\bar{H}_2}^{neg^2}(\bar{q}))_{\bar{c}} + (\bar{v}_{\bar{H}_2}^{neg^2}(\bar{q}))_{\bar{c}} + (\bar{\alpha}_{\bar{H}_2}^{pos^2}(\bar{q}))_{\bar{c}} + (\bar{\beta}_{\bar{H}_2}^{pos^2}(\bar{q}))_{\bar{c}} + (\bar{\alpha}_{\bar{H}_2}^{neg^2}(\bar{q}))_{\bar{c}} + (\bar{\beta}_{\bar{H}_2}^{neg^2}(\bar{q}))_{\bar{c}}}}{\bar{d}}$$

$$\leq 1$$

(ii) Holds obviously.

(iii) If $\bar{H}_1 = \bar{H}_2$, then

$$\begin{aligned}
(\bar{\mu}_{\bar{H}_1}^{pos}(\bar{q}))_{\bar{c}} = (\bar{\mu}_{\bar{H}_2}^{pos}(\bar{q}))_{\bar{c}} \quad &; \quad (\bar{v}_{\bar{H}_1}^{pos}(\bar{q}))_{\bar{c}} = (\bar{v}_{\bar{H}_2}^{pos}(\bar{q}))_{\bar{c}}; \\
(\bar{\mu}_{\bar{H}_1}^{neg}(\bar{q}))_{\bar{c}} = (\bar{\mu}_{\bar{H}_2}^{neg}(\bar{q}))_{\bar{c}} \quad &; \quad (\bar{v}_{\bar{H}_1}^{neg}(\bar{q}))_{\bar{c}} = (\bar{v}_{\bar{H}_2}^{neg}(\bar{q}))_{\bar{c}}; \\
(\bar{\alpha}_{\bar{H}_1}^{pos}(\bar{q}))_{\bar{c}} = (\bar{\alpha}_{\bar{H}_2}^{pos}(\bar{q}))_{\bar{c}} \quad &; \quad (\bar{\beta}_{\bar{H}_1}^{pos}(\bar{q}))_{\bar{c}} = (\bar{\beta}_{\bar{H}_2}^{pos}(\bar{q}))_{\bar{c}}; \\
(\bar{\alpha}_{\bar{H}_1}^{neg}(\bar{q}))_{\bar{c}} = (\bar{\alpha}_{\bar{H}_2}^{neg}(\bar{q}))_{\bar{c}} \quad &; \quad (\bar{\beta}_{\bar{H}_1}^{neg}(\bar{q}))_{\bar{c}} = (\bar{\beta}_{\bar{H}_2}^{neg}(\bar{q}))_{\bar{c}}.
\end{aligned}$$

Thus, $\overline{CS}_{\overline{BLDFH}}^1(\bar{H}_1, \bar{H}_2) = 1$. $\qquad\qquad\square$

**Theorem 3.3.** $\overline{CS}_{\overline{BLDFH}}^2(\bar{H}_1, \bar{H}_2)$ *holds the following properties:*

*(i)* $0 \leq \overline{CS}_{\overline{BLDFH}}^2(\bar{H}_1, \bar{H}_2) \leq 1$

*(ii)* $\overline{CS}_{\overline{BLDFH}}^2(\bar{H}_1, \bar{H}_2) = \overline{CS}_{\overline{BLDFH}}^2(\bar{H}_2, \bar{H}_1)$

*(iii)* $\bar{H}_1 = \bar{H}_2, \Leftrightarrow \overline{CS}_{\overline{BLDFH}}^2(\bar{H}_1, \bar{H}_2) = 1$.

*Proof.* (i) $0 \leq \overline{CS}_{\overline{BLDFH}}^1(\bar{H}_1, \bar{H}_2)$ is obvious.

To prove: $1 \geq \overline{CS}_{\overline{BLDFH}}^2(\bar{H}_1, \bar{H}_2)$

$$\Rightarrow \left[\begin{pmatrix} |(\bar{\mu}_{\bar{H}_1}^{pos}(\bar{q}))_{\bar{c}} - (\bar{\mu}_{\bar{H}_2}^{pos}(\bar{q}))_{\bar{c}}| \vee |(\bar{v}_{\bar{H}_1}^{pos}(\bar{q}))_{\bar{c}} - (\bar{v}_{\bar{H}_2}^{pos}(\bar{q}))_{\bar{c}}| \vee \\ |(\bar{\mu}_{\bar{H}_1}^{neg}(\bar{q}))_{\bar{c}} - (\bar{\mu}_{\bar{H}_2}^{neg}(\bar{q}))_{\bar{c}}| \vee |(\bar{v}_{\bar{H}_1}^{neg}(\bar{q}))_{\bar{c}} - (\bar{v}_{\bar{H}_2}^{neg}(\bar{q}))_{\bar{c}}| \vee \\ |(\bar{\alpha}_{\bar{H}_1}^{pos}(\bar{q}))_{\bar{c}} - (\bar{\alpha}_{\bar{H}_2}^{pos}(\bar{q}))_{\bar{c}}| \vee |(\bar{\beta}_{\bar{H}_1}^{pos}(\bar{q}))_{\bar{c}} - (\bar{\beta}_{\bar{H}_2}^{pos}(\bar{q}))_{\bar{c}}| \vee \\ |(\bar{\alpha}_{\bar{H}_1}^{neg}(\bar{q}))_{\bar{c}} - (\bar{\alpha}_{\bar{H}_2}^{neg}(\bar{q}))_{\bar{c}}| \vee |(\bar{\beta}_{\bar{H}_1}^{neg}(\bar{q}))_{\bar{c}} - (\bar{\beta}_{\bar{H}_2}^{neg}(\bar{q}))_{\bar{c}}| \end{pmatrix}\right] \leq 1$$

$$\Rightarrow \cos\left[\frac{\Pi}{4}\begin{pmatrix} |(\bar{\mu}_{\bar{H}_1}^{pos}(\bar{q}))_{\bar{c}} - (\bar{\mu}_{\bar{H}_2}^{pos}(\bar{q}))_{\bar{c}}| \vee |(\bar{v}_{\bar{H}_1}^{pos}(\bar{q}))_{\bar{c}} - (\bar{v}_{\bar{H}_2}^{pos}(\bar{q}))_{\bar{c}}| \vee \\ |(\bar{\mu}_{\bar{H}_1}^{neg}(\bar{q}))_{\bar{c}} - (\bar{\mu}_{\bar{H}_2}^{neg}(\bar{q}))_{\bar{c}}| \vee |(\bar{v}_{\bar{H}_1}^{neg}(\bar{q}))_{\bar{c}} - (\bar{v}_{\bar{H}_2}^{neg}(\bar{q}))_{\bar{c}}| \vee \\ |(\bar{\alpha}_{\bar{H}_1}^{pos}(\bar{q}))_{\bar{c}} - (\bar{\alpha}_{\bar{H}_2}^{pos}(\bar{q}))_{\bar{c}}| \vee |(\bar{\beta}_{\bar{H}_1}^{pos}(\bar{q}))_{\bar{c}} - (\bar{\beta}_{\bar{H}_2}^{pos}(\bar{q}))_{\bar{c}}| \vee \\ |(\bar{\alpha}_{\bar{H}_1}^{neg}(\bar{q}))_{\bar{c}} - (\bar{\alpha}_{\bar{H}_2}^{neg}(\bar{q}))_{\bar{c}}| \vee |(\bar{\beta}_{\bar{H}_1}^{neg}(\bar{q}))_{\bar{c}} - (\bar{\beta}_{\bar{H}_2}^{neg}(\bar{q}))_{\bar{c}}| \end{pmatrix}\right] \leq 1$$

$$\sum_{\bar{c}=1}^{\bar{w}} cos\left[\frac{\Pi}{4}\left(|(\bar{\mu}_{\bar{H}_1}^{pos}(\bar{q}))_{\bar{c}} - (\bar{\mu}_{\bar{H}_2}^{pos}(\bar{q}))_{\bar{c}}| \vee |(\bar{v}_{\bar{H}_1}^{pos}(\bar{q}))_{\bar{c}} - (\bar{v}_{\bar{H}_2}^{pos}(\bar{q}))_{\bar{c}}| \vee \right.\right.$$

$$|(\bar{\mu}_{\bar{H}_1}^{neg}(\bar{q}))_{\bar{c}} - (\bar{\mu}_{\bar{H}_2}^{neg}(\bar{q}))_{\bar{c}}| \vee |(\bar{v}_{\bar{H}_1}^{neg}(\bar{q}))_{\bar{c}} - (\bar{v}_{\bar{H}_2}^{neg}(\bar{q}))_{\bar{c}}| \vee$$

$$|(\bar{\alpha}_{\bar{H}_1}^{pos}(\bar{q}))_{\bar{c}} - (\bar{\alpha}_{\bar{H}_2}^{pos}(\bar{q}))_{\bar{c}}| \vee |(\bar{\beta}_{\bar{H}_1}^{pos}(\bar{q}))_{\bar{c}} - (\bar{\beta}_{\bar{H}_2}^{pos}(\bar{q}))_{\bar{c}}| \vee$$

$$\Rightarrow \frac{|(\bar{\alpha}_{\bar{H}_1}^{neg}(\bar{q}))_{\bar{c}} - (\bar{\alpha}_{\bar{H}_2}^{neg}(\bar{q}))_{\bar{c}}| \vee |(\bar{\beta}_{\bar{H}_1}^{neg}(\bar{q}))_{\bar{c}} - (\bar{\beta}_{\bar{H}_2}^{neg}(\bar{q}))_{\bar{c}}|)\Big]}{\bar{w}} \leq 1$$

Thus, $0 \leq \overline{CS}_{BLDFH}^2(\bar{H}_1, \bar{H}_2) \leq 1$

(ii) Holds obviously.

(iii) If $\bar{H}_1 = \bar{H}_2$, then

$$(\bar{\mu}_{\bar{H}_1}^{pos}(\bar{q}))_{\bar{c}} = (\bar{\mu}_{\bar{H}_2}^{pos}(\bar{q}))_{\bar{c}} \quad ; \quad (\bar{v}_{\bar{H}_1}^{pos}(\bar{q}))_{\bar{c}} = (\bar{v}_{\bar{H}_2}^{pos}(\bar{q}))_{\bar{c}};$$

$$(\bar{\mu}_{\bar{H}_1}^{neg}(\bar{q}))_{\bar{c}} = (\bar{\mu}_{\bar{H}_2}^{neg}(\bar{q}))_{\bar{c}} \quad ; \quad (\bar{v}_{\bar{H}_1}^{neg}(\bar{q}))_{\bar{c}} = (\bar{v}_{\bar{H}_2}^{neg}(\bar{q}))_{\bar{c}};$$

$$(\bar{\alpha}_{\bar{H}_1}^{pos}(\bar{q}))_{\bar{c}} = (\bar{\alpha}_{\bar{H}_2}^{pos}(\bar{q}))_{\bar{c}} \quad ; \quad (\bar{\beta}_{\bar{H}_1}^{pos}(\bar{q}))_{\bar{c}} = (\bar{\beta}_{\bar{H}_2}^{pos}(\bar{q}))_{\bar{c}};$$

$$(\bar{\alpha}_{\bar{H}_1}^{neg}(\bar{q}))_{\bar{c}} = (\bar{\alpha}_{\bar{H}_2}^{neg}(\bar{q}))_{\bar{c}} \quad ; \quad (\bar{\beta}_{\bar{H}_1}^{neg}(\bar{q}))_{\bar{c}} = (\bar{\beta}_{\bar{H}_2}^{neg}(\bar{q}))_{\bar{c}}.$$

Therefore,

$$|(\bar{\mu}_{\bar{H}_1}^{pos}(\bar{q}))_{\bar{c}} - (\bar{\mu}_{\bar{H}_2}^{pos}(\bar{q}))_{\bar{c}}| = 0 \quad ; \quad |(\bar{v}_{\bar{H}_1}^{pos}(\bar{q}))_{\bar{c}} - (\bar{v}_{\bar{H}_2}^{pos}(\bar{q}))_{\bar{c}}| = 0;$$

$$|(\bar{\mu}_{\bar{H}_1}^{neg}(\bar{q}))_{\bar{c}} - (\bar{\mu}_{\bar{H}_2}^{neg}(\bar{q}))_{\bar{c}}| = 0 \quad ; \quad |(\bar{v}_{\bar{H}_1}^{neg}(\bar{q}))_{\bar{c}} - (\bar{v}_{\bar{H}_2}^{neg}(\bar{q}))_{\bar{c}}| = 0;$$

$$|(\bar{\alpha}_{\bar{H}_1}^{pos}(\bar{q}))_{\bar{c}} - (\bar{\alpha}_{\bar{H}_2}^{pos}(\bar{q}))_{\bar{c}}| = 0 \quad ; \quad |(\bar{\beta}_{\bar{H}_1}^{pos}(\bar{q}))_{\bar{c}} - (\bar{\beta}_{\bar{H}_2}^{pos}(\bar{q}))_{\bar{c}}| = 0;$$

$$|(\bar{\alpha}_{\bar{H}_1}^{neg}(\bar{q}))_{\bar{c}} - (\bar{\alpha}_{\bar{H}_2}^{neg}(\bar{q}))_{\bar{c}}| = 0 \quad ; \quad |(\bar{\beta}_{\bar{H}_1}^{neg}(\bar{q}))_{\bar{c}} - (\bar{\beta}_{\bar{H}_2}^{neg}(\bar{q}))_{\bar{c}}| = 0.$$

Thus, $\overline{CS}_{BLDFH}^2(\bar{H}_1, \bar{H}_2) = 1.$ [$\because \cos(0) = 1$].

The Converse is also true and can be proved to be the same. $\qquad \square$

**Theorem 3.4.** $\overline{CS}_{BLDFH}^3(\bar{H}_1, \bar{H}_2)$ *holds the following properties:*

*(i)* $0 \leq \overline{CS}_{BLDFH}^3(\bar{H}_1, \bar{H}_2) \leq 1$
*(ii)* $\overline{CS}_{BLDFH}^3(\bar{H}_1, \bar{H}_2) = \overline{CS}_{BLDFH}^3(\bar{H}_2, \bar{H}_1)$
*(iii)* $\bar{H}_1 = \bar{H}_2, \Leftrightarrow \overline{CS}_{BLDFH}^3(\bar{H}_1, \bar{H}_2) = 1.$

*Proof.* (i) $0 \leq \overline{CS}_{BLDFH}^3(\bar{H}_1, \bar{H}_2)$ is obvious.
  To prove: $1 \geq \overline{CS}_{BLDFH}^3(\bar{H}_1, \bar{H}_2)$

$$\Rightarrow \left[\left(|(\bar{\mu}_{\bar{H}_1}^{pos}(\bar{q}))_{\bar{c}} - (\bar{\mu}_{\bar{H}_2}^{pos}(\bar{q}))_{\bar{c}}| \vee |(\bar{v}_{\bar{H}_1}^{pos}(\bar{q}))_{\bar{c}} - (\bar{v}_{\bar{H}_2}^{pos}(\bar{q}))_{\bar{c}}| \vee \right.\right.$$

$$|(\bar{\mu}_{\bar{H}_1}^{neg}(\bar{q}))_{\bar{c}} - (\bar{\mu}_{\bar{H}_2}^{neg}(\bar{q}))_{\bar{c}}| \vee |(\bar{v}_{\bar{H}_1}^{neg}(\bar{q}))_{\bar{c}} - (\bar{v}_{\bar{H}_2}^{neg}(\bar{q}))_{\bar{c}}| \vee$$

$$|(\bar{\alpha}_{\bar{H}_1}^{pos}(\bar{q}))_{\bar{c}} - (\bar{\alpha}_{\bar{H}_2}^{pos}(\bar{q}))_{\bar{c}}| \vee |(\bar{\beta}_{\bar{H}_1}^{pos}(\bar{q}))_{\bar{c}} - (\bar{\beta}_{\bar{H}_2}^{pos}(\bar{q}))_{\bar{c}}| \vee \qquad \leq 3$$

$$|(\bar{\alpha}_{\bar{H}_1}^{neg}(\bar{q}))_{\bar{c}} - (\bar{\alpha}_{\bar{H}_2}^{neg}(\bar{q}))_{\bar{c}}| \vee |(\bar{\beta}_{\bar{H}_1}^{neg}(\bar{q}))_{\bar{c}} - (\bar{\beta}_{\bar{H}_2}^{neg}(\bar{q}))_{\bar{c}}|\Big)\Big]$$

$$\Rightarrow \frac{1}{3}\left[\left(|(\bar{\mu}_{\bar{H}_1}^{pos}(\bar{q}))_{\bar{c}} - (\bar{\mu}_{\bar{H}_2}^{pos}(\bar{q}))_{\bar{c}}| \vee |(\bar{v}_{\bar{H}_1}^{pos}(\bar{q}))_{\bar{c}} - (\bar{v}_{\bar{H}_2}^{pos}(\bar{q}))_{\bar{c}}| \vee \right.\right.$$
$$|(\bar{\mu}_{\bar{H}_1}^{neg}(\bar{q}))_{\bar{c}} - (\bar{\mu}_{\bar{H}_2}^{neg}(\bar{q}))_{\bar{c}}| \vee |(\bar{v}_{\bar{H}_1}^{neg}(\bar{q}))_{\bar{c}} - (\bar{v}_{\bar{H}_2}^{neg}(\bar{q}))_{\bar{c}}| \vee$$
$$|(\bar{\alpha}_{\bar{H}_1}^{pos}(\bar{q}))_{\bar{c}} - (\bar{\alpha}_{\bar{H}_2}^{pos}(\bar{q}))_{\bar{c}}| \vee |(\bar{\beta}_{\bar{H}_1}^{pos}(\bar{q}))_{\bar{c}} - (\bar{\beta}_{\bar{H}_2}^{pos}(\bar{q}))_{\bar{c}}| \vee \quad \leq 1$$
$$\left.\left.|(\bar{\alpha}_{\bar{H}_1}^{neg}(\bar{q}))_{\bar{c}} - (\bar{\alpha}_{\bar{H}_2}^{neg}(\bar{q}))_{\bar{c}}| \vee |(\bar{\beta}_{\bar{H}_1}^{neg}(\bar{q}))_{\bar{c}} - (\bar{\beta}_{\bar{H}_2}^{neg}(\bar{q}))_{\bar{c}}|\right)\right]$$

$$\Rightarrow cos\left[\left(\frac{1}{3}\right)\left(\frac{\Pi}{2}\right)(|(\bar{\mu}_{\bar{H}_1}^{pos}(\bar{q}))_{\bar{c}} - (\bar{\mu}_{\bar{H}_2}^{pos}(\bar{q}))_{\bar{c}}| \vee |(\bar{v}_{\bar{H}_1}^{pos}(\bar{q}))_{\bar{c}} - (\bar{v}_{\bar{H}_2}^{pos}(\bar{q}))_{\bar{c}}| \vee. \right.$$
$$|(\bar{\mu}_{\bar{H}_1}^{neg}(\bar{q}))_{\bar{c}} - (\bar{\mu}_{\bar{H}_2}^{neg}(\bar{q}))_{\bar{c}}| \vee |(\bar{v}_{\bar{H}_1}^{neg}(\bar{q}))_{\bar{c}} - (\bar{v}_{\bar{H}_2}^{neg}(\bar{q}))_{\bar{c}}| \vee \quad \leq 1$$
$$|(\bar{\alpha}_{\bar{H}_1}^{pos}(\bar{q}))_{\bar{c}} - (\bar{\alpha}_{\bar{H}_2}^{pos}(\bar{q}))_{\bar{c}}| \vee |(\bar{\beta}_{\bar{H}_1}^{pos}(\bar{q}))_{\bar{c}} - (\bar{\beta}_{\bar{H}_2}^{pos}(\bar{q}))_{\bar{c}}| \vee$$
$$\left.|(\bar{\alpha}_{\bar{H}_1}^{neg}(\bar{q}))_{\bar{c}} - (\bar{\alpha}_{\bar{H}_2}^{neg}(\bar{q}))_{\bar{c}}| \vee |(\bar{\beta}_{\bar{H}_1}^{neg}(\bar{q}))_{\bar{c}} - (\bar{\beta}_{\bar{H}_2}^{neg}(\bar{q}))_{\bar{c}}|\right)\right]$$

$$\Rightarrow \frac{\sum_{\bar{c}=1}^{\bar{w}} cos\left[\frac{\Pi}{6}\left(|(\bar{\mu}_{\bar{H}_1}^{pos}(\bar{q}))_{\bar{c}} - (\bar{\mu}_{\bar{H}_2}^{pos}(\bar{q}))_{\bar{c}}| \vee |(\bar{v}_{\bar{H}_1}^{pos}(\bar{q}))_{\bar{c}} - (\bar{v}_{\bar{H}_2}^{pos}(\bar{q}))_{\bar{c}}| \vee \right.\right.}{\bar{w}}$$

with

$$|(\bar{\mu}_{\bar{H}_1}^{neg}(\bar{q}))_{\bar{c}} - (\bar{\mu}_{\bar{H}_2}^{neg}(\bar{q}))_{\bar{c}}| \vee |(\bar{v}_{\bar{H}_1}^{neg}(\bar{q}))_{\bar{c}} - (\bar{v}_{\bar{H}_2}^{neg}(\bar{q}))_{\bar{c}}| \vee$$
$$|(\bar{\alpha}_{\bar{H}_1}^{pos}(\bar{q}))_{\bar{c}} - (\bar{\alpha}_{\bar{H}_2}^{pos}(\bar{q}))_{\bar{c}}| \vee |(\bar{\beta}_{\bar{H}_1}^{pos}(\bar{q}))_{\bar{c}} - (\bar{\beta}_{\bar{H}_2}^{pos}(\bar{q}))_{\bar{c}}| \vee$$
$$\left.\left.|(\bar{\alpha}_{\bar{H}_1}^{neg}(\bar{q}))_{\bar{c}} - (\bar{\alpha}_{\bar{H}_2}^{neg}(\bar{q}))_{\bar{c}}| \vee |(\bar{\beta}_{\bar{H}_1}^{neg}(\bar{q}))_{\bar{c}} - (\bar{\beta}_{\bar{H}_2}^{neg}(\bar{q}))_{\bar{c}}|\right)\right] \leq 1$$

Thus, $0 \leq \overline{CS}_{\overline{BLDFH}}^3(\bar{H}_1, \bar{H}_2) \leq 1$

(ii) and (iii) follows from theorem: 4.3. □

**Definition 3.5.** *Let the alternative set,* $\bar{X} = \{\bar{X}_1, \bar{X}_2, \ldots, \bar{X}_\gamma\}$ *and two BLDFHS be*

$$\bar{H}_1 = \langle \bar{i}, (\bar{q}, \{\bar{\mu}_{\bar{H}_1}^{pos}(\bar{q}), \bar{v}_{\bar{H}_1}^{pos}(\bar{q}), \bar{\mu}_{\bar{H}_1}^{neg}(\bar{q}), \bar{v}_{\bar{H}_1}^{neg}(\bar{q})\}, \{\bar{\alpha}_{\bar{H}_1}^{pos}(\bar{q}), \bar{\beta}_{\bar{H}_1}^{pos}(\bar{q}), \bar{\alpha}_{\bar{H}_1}^{neg}(\bar{q}), \bar{\beta}_{\bar{H}_1}^{neg}(\bar{q})\})\rangle,$$

*and*

$$\bar{H}_2 = \langle \bar{i}, (\bar{q}, \{\bar{\mu}_{\bar{H}_2}^{pos}(\bar{q}), \bar{v}_{\bar{H}_2}^{pos}(\bar{q}), \bar{\mu}_{\bar{H}_2}^{neg}(\bar{q}), \bar{v}_{\bar{H}_2}^{neg}(\bar{q})\}, \{\bar{\alpha}_{\bar{H}_2}^{pos}(\bar{q}), \bar{\beta}_{\bar{H}_2}^{pos}(\bar{q}), \bar{\alpha}_{\bar{H}_2}^{neg}(\bar{q}), \bar{\beta}_{\bar{H}_2}^{neg}(\bar{q})\})\rangle.$$

*Then the cotangent similarity measure between* $\bar{H}_1$ *and* $\bar{H}_2$ *with the usage of arithmetic mean is offered as:*

$$\overline{CS}_{\overline{BLDFH}}^4(\bar{H}_1, \bar{H}_2) =$$

$$\frac{\sum_{\bar{c}=1}^{\bar{w}} cot\left[\frac{\Pi}{4} + \frac{\Pi}{4}\left(|(\bar{\mu}_{\bar{H}_1}^{pos}(\bar{q}))_{\bar{c}} - (\bar{\mu}_{\bar{H}_2}^{pos}(\bar{q}))_{\bar{c}}| \vee |(\bar{v}_{\bar{H}_1}^{pos}(\bar{q}))_{\bar{c}} - (\bar{v}_{\bar{H}_2}^{pos}(\bar{q}))_{\bar{c}}| \vee \right.\right.}{\bar{w}}$$

$$|(\bar{\mu}_{\bar{H}_1}^{neg}(\bar{q}))_{\bar{c}} - (\bar{\mu}_{\bar{H}_2}^{neg}(\bar{q}))_{\bar{c}}| \vee |(\bar{v}_{\bar{H}_1}^{neg}(\bar{q}))_{\bar{c}} - (\bar{v}_{\bar{H}_2}^{neg}(\bar{q}))_{\bar{c}}| \vee$$
$$|(\bar{\alpha}_{\bar{H}_1}^{pos}(\bar{q}))_{\bar{c}} - (\bar{\alpha}_{\bar{H}_2}^{pos}(\bar{q}))_{\bar{c}}| \vee |(\bar{\beta}_{\bar{H}_1}^{pos}(\bar{q}))_{\bar{c}} - (\bar{\beta}_{\bar{H}_2}^{pos}(\bar{q}))_{\bar{c}}| \vee$$
$$\left.\left.|(\bar{\alpha}_{\bar{H}_1}^{neg}(\bar{q}))_{\bar{c}} - (\bar{\alpha}_{\bar{H}_2}^{neg}(\bar{q}))_{\bar{c}}| \vee |(\bar{\beta}_{\bar{H}_1}^{neg}(\bar{q}))_{\bar{c}} - (\bar{\beta}_{\bar{H}_2}^{neg}(\bar{q}))_{\bar{c}}|\right)\right]}{} \tag{6}$$

$$\overline{CS}^3_{BLDFH}(\bar{H}_1, \bar{H}_2) =$$

$$\sum_{\bar{c}=1}^{\bar{w}} cot \left[ \frac{\Pi}{4} + \frac{\Pi}{12} \left( |(\bar{\mu}^{pos}_{\bar{H}_1}(\bar{q}))_{\bar{c}} - (\bar{\mu}^{pos}_{\bar{H}_2}(\bar{q}))_{\bar{c}}| \vee |(\bar{v}^{pos}_{\bar{H}_1}(\bar{q}))_{\bar{c}} - (\bar{v}^{pos}_{\bar{H}_2}(\bar{q}))_{\bar{c}}| \vee \right. \right.$$

$$|(\bar{\mu}^{neg}_{\bar{H}_1}(\bar{q}))_{\bar{c}} - (\bar{\mu}^{neg}_{\bar{H}_2}(\bar{q}))_{\bar{c}}| \vee |(\bar{v}^{neg}_{\bar{H}_1}(\bar{q}))_{\bar{c}} - (\bar{v}^{neg}_{\bar{H}_2}(\bar{q}))_{\bar{c}}| \vee$$

$$|(\bar{\alpha}^{pos}_{\bar{H}_1}(\bar{q}))_{\bar{c}} - (\bar{\alpha}^{pos}_{\bar{H}_2}(\bar{q}))_{\bar{c}}| \vee |(\bar{\beta}^{pos}_{\bar{H}_1}(\bar{q}))_{\bar{c}} - (\bar{\beta}^{pos}_{\bar{H}_2}(\bar{q}))_{\bar{c}}| \vee$$

$$\left. \left. \frac{|(\bar{\alpha}^{neg}_{\bar{H}_1}(\bar{q}))_{\bar{c}} - (\bar{\alpha}^{neg}_{\bar{H}_2}(\bar{q}))_{\bar{c}}| \vee |(\bar{\beta}^{neg}_{\bar{H}_1}(\bar{q}))_{\bar{c}} - (\bar{\beta}^{neg}_{\bar{H}_2}(\bar{q}))_{\bar{c}}|}{\bar{w}} \right) \right] \tag{7}$$

**Proposition 3.6.** $\overline{CS}^4_{BLDFH}(\bar{H}_1, \bar{H}_2)$ & $\overline{CS}^5_{BLDFH}(\bar{H}_1, \bar{H}_2)$ *satisfies all the above properties.*

*Proof.* Using the theorems 4.3 & 4.4, it can be proved. □

**Definition 3.7.** *Weighted cosine & cotangent similarities are give below:*

(i) $\overline{WCS}^1_{BLDFH}(\overline{H}_1, \overline{H}_2) =$

$$\sum_{\bar{c}=1}^{\bar{d}} \overline{W_n} \frac{\begin{pmatrix} \left( (\overline{\mu}^{pos}_{\overline{H}_1}(\bar{q}))_{\bar{c}} (\overline{\mu}^{pos}_{\overline{H}_2}(\bar{q}))_{\bar{c}} \right) + \left( (\overline{v}^{pos}_{\overline{H}_1}(\bar{q}))_{\bar{c}} (\overline{v}^{pos}_{\overline{H}_2}(\bar{q}))_{\bar{c}} \right) + \left( (\overline{\mu}^{neg}_{\overline{H}_1}(\bar{q}))_{\bar{c}} (\overline{\mu}^{neg}_{\overline{H}_2}(\bar{q}))_{\bar{c}} \right) + \left( (\overline{v}^{neg}_{\overline{H}_1}(\bar{q}))_{\bar{c}} (\overline{v}^{neg}_{\overline{H}_2}(\bar{q}))_{\bar{c}} \right), \\ \left( (\overline{\alpha}^{pos}_{\overline{H}_1}(\bar{q}))_{\bar{c}} (\overline{\alpha}^{pos}_{\overline{H}_2}(\bar{q}))_{\bar{c}} \right) + \left( (\overline{\beta}^{pos}_{\overline{H}_1}(\bar{q}))_{\bar{c}} (\overline{\beta}^{pos}_{\overline{H}_2}(\bar{q}))_{\bar{c}} \right) + \left( (\overline{\alpha}^{neg}_{\overline{H}_1}(\bar{q}))_{\bar{c}} (\overline{\alpha}^{neg}_{\overline{H}_2}(\bar{q}))_{\bar{c}} \right) + \left( (\overline{\beta}^{neg}_{\overline{H}_1}(\bar{q}))_{\bar{c}} (\overline{\beta}^{neg}_{\overline{H}_2}(\bar{q}))_{\bar{c}} \right) \end{pmatrix}}{\dfrac{\sqrt{(\overline{\mu}^{pos^2}_{\overline{H}_1}(\bar{q}))_{\bar{c}} + (\overline{v}^{pos^2}_{\overline{H}_1}(\bar{q}))_{\bar{c}} + (\overline{\mu}^{neg^2}_{\overline{H}_1}(\bar{q}))_{\bar{c}} + (\overline{v}^{neg^2}_{\overline{H}_1}(\bar{q}))_{\bar{c}} + (\overline{\alpha}^{pos^2}_{\overline{H}_1}(\bar{q}))_{\bar{c}} + (\overline{\beta}^{pos^2}_{\overline{H}_1}(\bar{q}))_{\bar{c}} + (\overline{\alpha}^{neg^2}_{\overline{H}_1}(\bar{q}))_{\bar{c}} + (\overline{\beta}^{neg^2}_{\overline{H}_1}(\bar{q}))_{\bar{c}}}}{2} \cdot \dfrac{\sqrt{(\overline{\mu}^{pos^2}_{\overline{H}_2}(\bar{q}))_{\bar{c}} + (\overline{v}^{pos^2}_{\overline{H}_2}(\bar{q}))_{\bar{c}} + (\overline{\mu}^{neg^2}_{\overline{H}_2}(\bar{q}))_{\bar{c}} + (\overline{v}^{neg^2}_{\overline{H}_2}(\bar{q}))_{\bar{c}} + (\overline{\alpha}^{pos^2}_{\overline{H}_2}(\bar{q}))_{\bar{c}} + (\overline{\beta}^{pos^2}_{\overline{H}_2}(\bar{q}))_{\bar{c}} + (\overline{\alpha}^{neg^2}_{\overline{H}_2}(\bar{q}))_{\bar{c}} + (\overline{\beta}^{neg^2}_{\overline{H}_2}(\bar{q}))_{\bar{c}}}}{\overline{d}}}$$

(ii) $\overline{WCS}^2_{BLDFH}(\bar{H}_1, \bar{H}_2) =$

$$\sum_{\bar{c}=1}^{\bar{w}} \overline{W_n} cos \left[ \frac{\Pi}{4} \left( |(\bar{\mu}^{pos}_{\bar{H}_1}(\bar{q}))_{\bar{c}} - (\bar{\mu}^{pos}_{\bar{H}_2}(\bar{q}))_{\bar{c}}| \vee |(\bar{v}^{pos}_{\bar{H}_1}(\bar{q}))_{\bar{c}} - (\bar{v}^{pos}_{\bar{H}_2}(\bar{q}))_{\bar{c}}| \vee \right. \right.$$

$$|(\bar{\mu}^{neg}_{\bar{H}_1}(\bar{q}))_{\bar{c}} - (\bar{\mu}^{neg}_{\bar{H}_2}(\bar{q}))_{\bar{c}}| \vee |(\bar{v}^{neg}_{\bar{H}_1}(\bar{q}))_{\bar{c}} - (\bar{v}^{neg}_{\bar{H}_2}(\bar{q}))_{\bar{c}}| \vee$$

$$|(\bar{\alpha}^{pos}_{\bar{H}_1}(\bar{q}))_{\bar{c}} - (\bar{\alpha}^{pos}_{\bar{H}_2}(\bar{q}))_{\bar{c}}| \vee |(\bar{\beta}^{pos}_{\bar{H}_1}(\bar{q}))_{\bar{c}} - (\bar{\beta}^{pos}_{\bar{H}_2}(\bar{q}))_{\bar{c}}| \vee$$

$$\left. \left. \frac{|(\bar{\alpha}^{neg}_{\bar{H}_1}(\bar{q}))_{\bar{c}} - (\bar{\alpha}^{neg}_{\bar{H}_2}(\bar{q}))_{\bar{c}}| \vee |(\bar{\beta}^{neg}_{\bar{H}_1}(\bar{q}))_{\bar{c}} - (\bar{\beta}^{neg}_{\bar{H}_2}(\bar{q}))_{\bar{c}}|}{\bar{w}} \right) \right]$$

(iii) $\overline{WCS}^3_{BLDFH}(\bar{H}_1, \bar{H}_2) =$

$$\sum_{\bar{c}=1}^{\bar{w}} \overline{W_n} cos \left[ \frac{\Pi}{6} \left( |(\bar{\mu}^{pos}_{\bar{H}_1}(\bar{q}))_{\bar{c}} - (\bar{\mu}^{pos}_{\bar{H}_2}(\bar{q}))_{\bar{c}}| \vee |(\bar{v}^{pos}_{\bar{H}_1}(\bar{q}))_{\bar{c}} - (\bar{v}^{pos}_{\bar{H}_2}(\bar{q}))_{\bar{c}}| \vee \right. \right.$$

$$|(\bar{\mu}^{neg}_{\bar{H}_1}(\bar{q}))_{\bar{c}} - (\bar{\mu}^{neg}_{\bar{H}_2}(\bar{q}))_{\bar{c}}| \vee |(\bar{v}^{neg}_{\bar{H}_1}(\bar{q}))_{\bar{c}} - (\bar{v}^{neg}_{\bar{H}_2}(\bar{q}))_{\bar{c}}| \vee$$

$$|(\bar{\alpha}^{pos}_{\bar{H}_1}(\bar{q}))_{\bar{c}} - (\bar{\alpha}^{pos}_{\bar{H}_2}(\bar{q}))_{\bar{c}}| \vee |(\bar{\beta}^{pos}_{\bar{H}_1}(\bar{q}))_{\bar{c}} - (\bar{\beta}^{pos}_{\bar{H}_2}(\bar{q}))_{\bar{c}}| \vee$$

$$\left. \left. \frac{|(\bar{\alpha}^{neg}_{\bar{H}_1}(\bar{q}))_{\bar{c}} - (\bar{\alpha}^{neg}_{\bar{H}_2}(\bar{q}))_{\bar{c}}| \vee |(\bar{\beta}^{neg}_{\bar{H}_1}(\bar{q}))_{\bar{c}} - (\bar{\beta}^{neg}_{\bar{H}_2}(\bar{q}))_{\bar{c}}|}{\bar{w}} \right) \right]$$

*(iv)* $\overline{WCS}^4_{BLDFH}(\bar{H}_1, \bar{H}_2) =$

$$\sum_{\bar{c}=1}^{\bar{w}} \overline{W_n} \cot\left[\frac{\Pi}{4} + \frac{\Pi}{4}\left(|(\bar{\mu}^{pos}_{\bar{H}_1}(\bar{q}))_{\bar{c}} - (\bar{\mu}^{pos}_{\bar{H}_2}(\bar{q}))_{\bar{c}}| \vee |(\bar{v}^{pos}_{\bar{H}_1}(\bar{q}))_{\bar{c}} - (\bar{v}^{pos}_{\bar{H}_2}(\bar{q}))_{\bar{c}}| \vee\right.\right.$$
$$|(\bar{\mu}^{neg}_{\bar{H}_1}(\bar{q}))_{\bar{c}} - (\bar{\mu}^{neg}_{\bar{H}_2}(\bar{q}))_{\bar{c}}| \vee |(\bar{v}^{neg}_{\bar{H}_1}(\bar{q}))_{\bar{c}} - (\bar{v}^{neg}_{\bar{H}_2}(\bar{q}))_{\bar{c}}| \vee$$
$$|(\bar{\alpha}^{pos}_{\bar{H}_1}(\bar{q}))_{\bar{c}} - (\bar{\alpha}^{pos}_{\bar{H}_2}(\bar{q}))_{\bar{c}}| \vee |(\bar{\beta}^{pos}_{\bar{H}_1}(\bar{q}))_{\bar{c}} - (\bar{\beta}^{pos}_{\bar{H}_2}(\bar{q}))_{\bar{c}}| \vee$$
$$\left.\left.\frac{|(\bar{\alpha}^{neg}_{\bar{H}_1}(\bar{q}))_{\bar{c}} - (\bar{\alpha}^{neg}_{\bar{H}_2}(\bar{q}))_{\bar{c}}| \vee |(\bar{\beta}^{neg}_{\bar{H}_1}(\bar{q}))_{\bar{c}} - (\bar{\beta}^{neg}_{\bar{H}_2}(\bar{q}))_{\bar{c}}|\right)\right]}{\bar{w}}$$

*(v)* $\overline{WCS}^5_{BLDFH}(\bar{H}_1, \bar{H}_2) =$

$$\sum_{\bar{c}=1}^{\bar{w}} \overline{W_n} \cos\left[\frac{\Pi}{4} + \frac{\Pi}{12}\left(|(\bar{\mu}^{pos}_{\bar{H}_1}(\bar{q}))_{\bar{c}} - (\bar{\mu}^{pos}_{\bar{H}_2}(\bar{q}))_{\bar{c}}| \vee |(\bar{v}^{pos}_{\bar{H}_1}(\bar{q}))_{\bar{c}} - (\bar{v}^{pos}_{\bar{H}_2}(\bar{q}))_{\bar{c}}| \vee\right.\right.$$
$$|(\bar{\mu}^{neg}_{\bar{H}_1}(\bar{q}))_{\bar{c}} - (\bar{\mu}^{neg}_{\bar{H}_2}(\bar{q}))_{\bar{c}}| \vee |(\bar{v}^{neg}_{\bar{H}_1}(\bar{q}))_{\bar{c}} - (\bar{v}^{neg}_{\bar{H}_2}(\bar{q}))_{\bar{c}}| \vee$$
$$|(\bar{\alpha}^{pos}_{\bar{H}_1}(\bar{q}))_{\bar{c}} - (\bar{\alpha}^{pos}_{\bar{H}_2}(\bar{q}))_{\bar{c}}| \vee |(\bar{\beta}^{pos}_{\bar{H}_1}(\bar{q}))_{\bar{c}} - (\bar{\beta}^{pos}_{\bar{H}_2}(\bar{q}))_{\bar{c}}| \vee$$
$$\left.\left.\frac{|(\bar{\alpha}^{neg}_{\bar{H}_1}(\bar{q}))_{\bar{c}} - (\bar{\alpha}^{neg}_{\bar{H}_2}(\bar{q}))_{\bar{c}}| \vee |(\bar{\beta}^{neg}_{\bar{H}_1}(\bar{q}))_{\bar{c}} - (\bar{\beta}^{neg}_{\bar{H}_2}(\bar{q}))_{\bar{c}}|\right)\right]}{\bar{w}}$$

*Here, $1 \geq \overline{W_1}, \overline{W_2}, \ldots, \overline{W_n} \geq 0$ are the weights with $\sum_{\bar{s}=1}^{\bar{n}} \overline{W_s} = 1$.*

## BLDFH TRIGONOMETRIC SIMILARITY ALGORITHM

An algorithmic approach for MADM problems is given as follows:

**Step 1:** Construct BLDFHS based on decision-makers choices.

**Step 2:** Using the proposed definitions: 3.1 (or) 3.5, find the similarity measures.

**Step 3:** Rank the alternatives based on the highest similarity value obtained.

A diagrammatic flowchart is given below (Fig. 1).

## CASE STUDY

Dengue is a mosquito-borne viral infection caused by the dengue virus (DENV), which belongs to the *Flavivirus* genus. It is transmitted primarily by *Aedes aegypti* and *Aedes albopictus* mosquitoes. Dengue is endemic in tropical and subtropical regions worldwide, with millions of cases reported annually. The disease manifests in a broad clinical spectrum, ranging from mild febrile illness to severe complications such as dengue haemorrhagic fever (DHF) and DSS.

**Types of dengue:**
Dengue infection is classified into the following categories based on severity:

- **Dengue fever (DF):**
  * The mildest form, characterized by high fever, severe headaches, retro-orbital pain, joint and muscle pain, and skin rashes.
  * Symptoms typically last 5–7 days and resolve without severe complications.

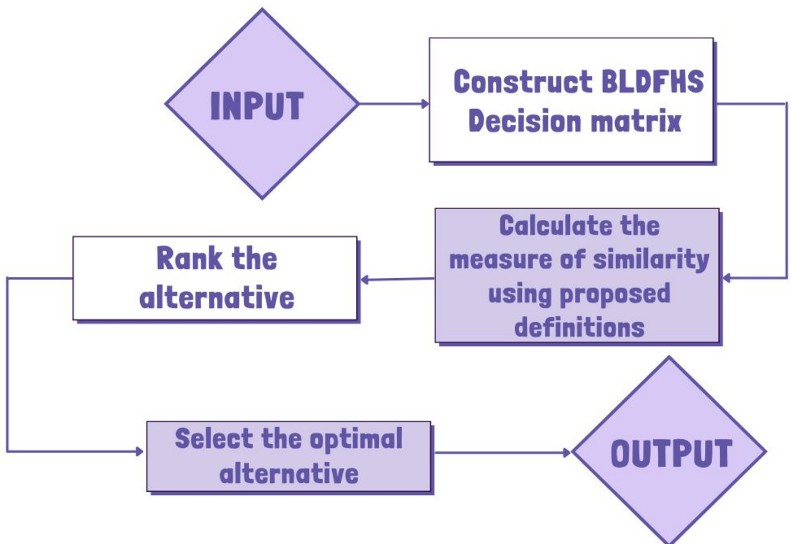

**Figure 1 Algorithmic flowchart of the BLDFHS trigonometric similarity measure.**

- **Dengue haemorrhagic fever:**
  * A severe form involving increased vascular permeability, plasma leakage, and haemorrhagic manifestations.
  * Symptoms include persistent vomiting, severe abdominal pain, gum or nose bleeding, easy bruising, and petechiae (small red spots under the skin due to bleeding).
  * If left untreated, DHF can progress to life-threatening shock.
- **Dengue shock syndrome:**
  * The most severe form of dengue, where plasma leakage leads to critical hypovolemic shock.
  * Symptoms include profound hypotension (low blood pressure), circulatory failure, multiple organ dysfunction, and severe haemorrhagic manifestations.
  * DSS is considered the most dangerous form due to its high mortality rate and delayed detection.

**Dengue shock syndrome: the most dangerous form**

DSS occurs when excessive plasma leakage results in critical hypotension, impairing oxygen delivery to vital organs. This condition takes longer to detect because early symptoms resemble those of classic dengue fever, making timely intervention challenging.

**Key symptoms of DSS**

1. **Hypotension (low blood pressure):**
   ○ Defined as a systolic blood pressure below 90 mmHg in adults and below 70 mmHg in children.

- Symptoms include dizziness, cold clammy skin, rapid weak pulse, and confusion.
- If untreated, it leads to circulatory collapse and shock.

2. **Haemorrhagic manifestations:**
   - Severe bleeding due to platelet dysfunction and endothelial damage that includes, petechiae (small red spots), purpura (large purple spots), and hematemesis (vomiting blood). Severe cases of these may result in intracranial haemorrhage, which is life-threatening.

3. **Thrombocytopenia (low platelet count):**
   - Platelet count drops below 100,000 cells/$\mu$L (normal range: 150,000–450,000 cells/$\mu$L).
   - Increases the risk of spontaneous bleeding and haemorrhagic complications.
   - Severe thrombocytopenia (platelets below 20,000 cells/$\mu$L) may require platelet transfusion.

Dengue, particularly DSS, poses a significant public health challenge due to its rapid progression and delayed detection. Early recognition of symptoms such as hypotension, haemorrhagic manifestations, and thrombocytopenia is crucial for timely intervention. Prompt fluid resuscitation, close monitoring, and supportive care play a vital role in reducing mortality and improving patient outcomes. Continued efforts in vector control, early diagnosis, and public awareness are essential in mitigating the impact of dengue worldwide.

To address this challenge, the BLDFHS trigonometric similarity measure serves as a diagnostic tool. The universal set consists of three patients $\bar{X} = \{\bar{X}_1, \bar{X}_2, \bar{X}_3\}$, each exhibiting varying degrees of $\bar{Y}_{\bar{1}}$ = hypotension, $\bar{Y}_{\bar{2}}$ = hemorrhagic manifestations, and $\bar{Y}_{\bar{3}}$ = thrombocytopenia, along with their respective attribute values $\bar{M}_{\bar{Y}_1}$ = {mild, moderate, severe}, $\bar{M}_{\bar{Y}_2}$ = {petechiae, purpura, hematemesis}, $\bar{M}_{\bar{Y}_3}$ = {100,000–150,000, 50,000–100,000, <50,000}. Let $\bar{\upsilon}_1$ = {moderate, purpura, 50,000–100,000}, $\bar{\upsilon}_2$ = {mild, petechiae, <50,000}, $\bar{\upsilon}_3$ = {severe, haematemesis, <50,000}. The values of $\bar{\mu}^{pos}, \bar{\upsilon}^{pos}, \bar{\mu}^{neg}, \bar{\upsilon}^{neg}$ represents how much the patient is more probable, not more probable, less probable, and not less probable to have DSS respectively. It is evaluated independently by multiple haematologists, reflecting realistic clinical variability in symptom assessment. This aligns with the nature of fuzzy hypersoft decision models, which are designed to accommodate subjective and uncertain expert input. By computing the similarity measure based on the proposed BLDFHS framework, the diagnostic system will help the haematologist compare patient symptoms against known, more specific criteria for symptomatic DSS.

The proposed method will facilitate:

(i) Rapid and accurate identification of DSS cases.

(ii) Reduction of misdiagnosis and unnecessary delays in treatment.

(iii) A systematic decision-support mechanism to assist in clinical evaluations.

(iv) Lower mortality rates by enabling timely and appropriate medical intervention.

**Table 2  Decision grid for DSS symptom assessment.**

|  | BLDFHS values |
|---|---|
| $\bar{v}_1$ | {(0.8, 0.3, −0.5, −0.7) (0.8, 0.2, −0.4, −0.5)} |
| $\bar{v}_2$ | {(0.7, 0.4, −0.4, −0.8) (0.4, 0.5, −0.2, −0.7)} |
| $\bar{v}_3$ | {(0.8, 0.2, −0.4, −0.5) (0.4, 0.3, −0.3, −0.7)} |

**Table 3  Decision grid for $\bar{X}_1$ symptom assessment.**

|  | BLDFHS values |
|---|---|
| $\bar{v}_1$ | {(0.4, 0.4, −0.5, −0.8) (0.7, 0.3, −0.4, −0.2)} |
| $\bar{v}_2$ | {(0.6, 0.3, −0.4, −0.4) (0.5, 0.3, −0.2, −0.6)} |
| $\bar{v}_3$ | {(0.8, 0.2, −0.3, −0.4) (0.4, 0.3, −0.6, −0.8)} |

**Table 4  Decision grid for $\bar{X}_2$ symptom assessment.**

|  | BLDFHS values |
|---|---|
| $\bar{v}_1$ | {(0.7, 0.5, −0.4, −0.6) (0.6, 0.2, −0.5, −0.4)} |
| $\bar{v}_2$ | {(0.6, 0.2, −0.5, −0.8) (0.5, 0.5, −0.1, −0.7)} |
| $\bar{v}_3$ | {(0.8, 0.1, −0.5, −0.4) (0.4, 0.2, −0.3, −0.6)} |

**Table 5  Decision grid for $\bar{X}_3$ symptom assessment.**

|  | BLDFHS values |
|---|---|
| $\bar{v}_1$ | {(0.4, 0.3, −0.4, −0.7) (0.8, 0.2, −0.3, −0.1)} |
| $\bar{v}_2$ | {(0.5, 0.2, −0.4, −0.5) (0.4, 0.3, −0.2, −0.5)} |
| $\bar{v}_3$ | {(0.5, 0.3, −0.4, −0.5) (0.5, 0.3, −0.2, −0.8)} |

## SOLUTION TO THE PROBLEM BASED ON BLDFH TRIGONOMETRIC SIMILARITY ALGORITHM

**Step 1**: Table 2 is the BLDFHS construction of DSS symptoms. Tables 3, 4, and 5 are the construction of BLDFHS for each patient with specific symptoms based on the haematologist's decision.

**Step 2**: Using Eq. (4), compute the similarity measure as follows:

$$\overline{CS}^2_{BLDFH}(DSS, \bar{X}_1)$$
$$= \frac{\frac{1}{\sqrt{2}}(0.4) + \frac{1}{\sqrt{2}}(0.2) + \frac{1}{\sqrt{2}}(0.3)}{3}$$
$$= \frac{0.2 + 0.1 + 0.2}{3}$$
$$= \frac{0.5}{3}$$
$$= 0.1$$

**Table 6 Comparative analysis with other proposed similarity measures.**

| BLDFHS Trigonometric similarity measure | Ranking |
| --- | --- |
| $\overline{CS^2_{BLDFH}}$ | $\bar{X}_3(0.2) > \bar{X}_1(0.1) > \bar{X}_2(0.09)$ |
| $\overline{CS^3_{BLDFH}}$ | $\bar{X}_3(0.233) > \bar{X}_1(0.2) > \bar{X}_2(0.133)$ |
| $\overline{CS^4_{BLDFH}}$ | $\bar{X}_3(0.66) > \bar{X}_1(0.6) = \bar{X}_2(0.6)$ |

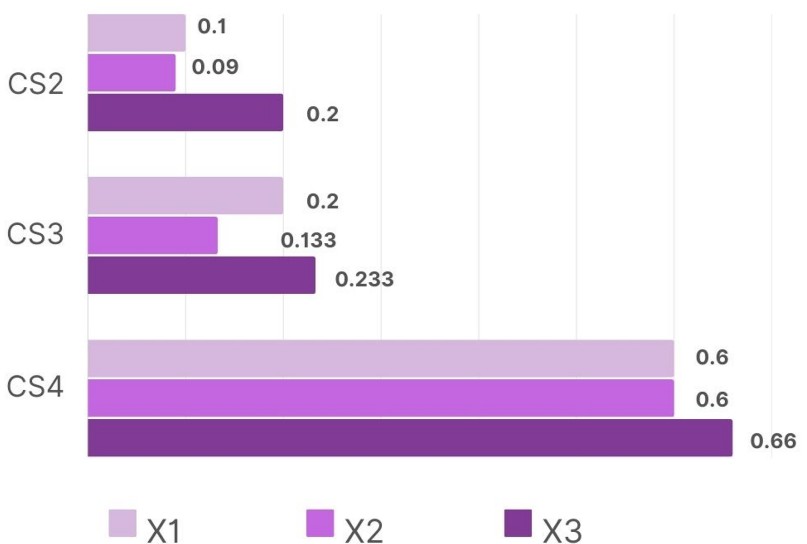

**Figure 2 Graphical comparison of other proposed similarity measures in DSS diagnosis.**

Likewise,

$$\overline{CS^2_{BLDFH}}(DSS, \bar{X}_2) = 0.09$$

$$\overline{CS^2_{BLDFH}}(DSS, \bar{X}_3) = 0.2.$$

**Step: 3** Based on **Step: 2**, $\bar{X}_3 > \bar{X}_1 > \bar{X}_2$.
Patient 3, $\bar{X}_3$ is more likely to have DSS.

## COMPARISON

### Comparative evaluation with other proposed cosine and cotangent similarity measures

The results obtained using the proposed BLDFHS Trigonometric Similarity Measure are compared with other proposed cosine and cotangent similarity measures to evaluate its effectiveness in diagnosing Dengue Shock Syndrome (DSS).

The Table 6 demonstrates the accuracy and reliability of the proposed BLDFHS Trigonometric Similarity Measure in diagnosing Dengue Shock Syndrome (DSS) and is represented in the form of a graph as below (Fig. 2).

Table 7 Comparative analysis with existing similarity measures.

| Fuzzy set | Trigonometric similarity measure | Ranking |
|---|---|---|
| IFHS (*Jafar et al., 2024*) | $\overline{CS}^2_{IFH}$ | $\bar{X}_3 > \bar{X}_1 = \bar{X}_2$ |
| | $\overline{CS}^3_{IFH}$ | $\bar{X}_3 > \bar{X}_1 > \bar{X}_2$ |
| | $\overline{CS}^4_{IFH}$ | $\bar{X}_3 > \bar{X}_1 = \bar{X}_2$ |
| LDFS (*Mohammad, Abdullah & Al-Shomrani, 2022*) | $\overline{CS}^2_{LDF}$ | $\bar{X}_3 > \bar{X}_1 > \bar{X}_2$ |
| | $\overline{CS}^4_{LDF}$ | $\bar{X}_3 > \bar{X}_1 > \bar{X}_2$ |
| | $\overline{CS}^4_{LDF}$ | $\bar{X}_3 > \bar{X}_1 > \bar{X}_2$ |

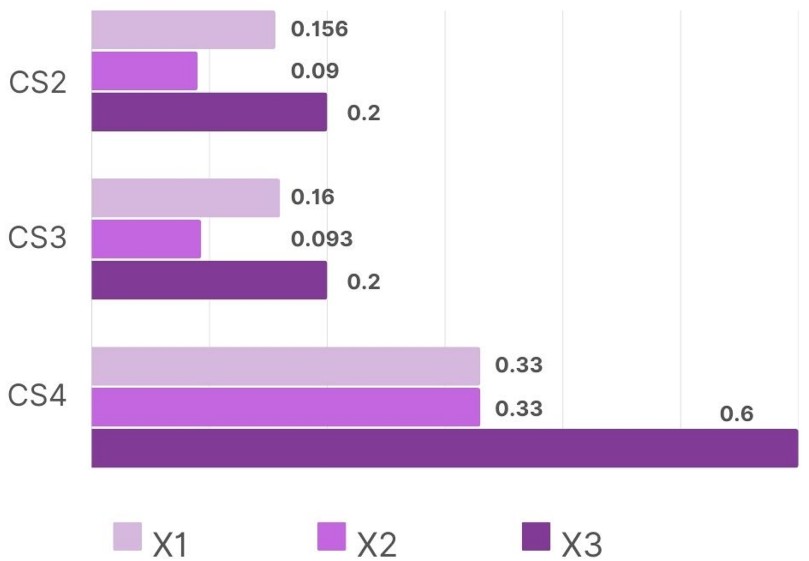

Figure 3 Graphical comparison with IFHS in DSS diagnosis.

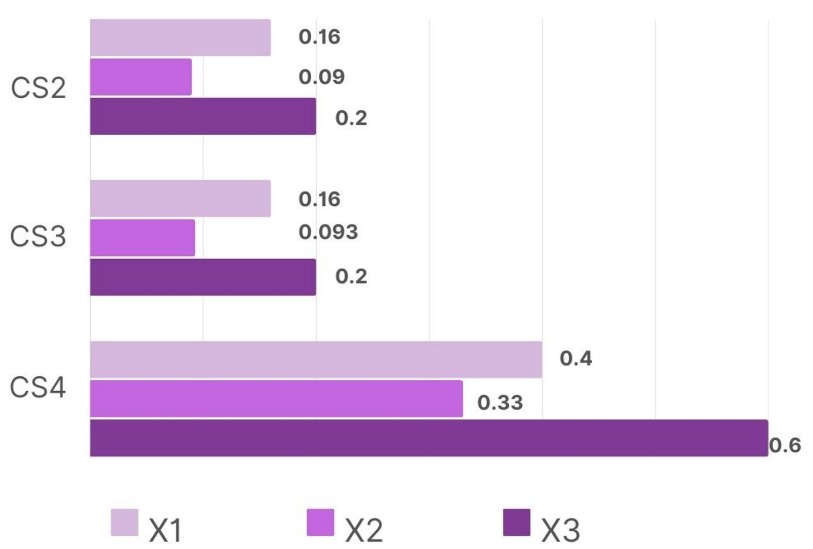

Figure 4 Graphical comparison with LDFS in DSS diagnosis.

## Comparative evaluation with other existing cosine and cotangent similarity measures

The results obtained using the proposed BLDFHS Trigonometric Similarity Measure are compared with other existing cosine and cotangent similarity measures like Intuitionistic Fuzzy Hypersoft Set (IFHS) and Linear Diophantine Fuzzy Set (LDFS) to evaluate its effectiveness in diagnosing Dengue Shock Syndrome (DSS).

The Table 7 demonstrates the accuracy and reliability of the proposed model with existing ones in diagnosing Dengue Shock Syndrome (DSS) and is represented in the form of a graph as below (Figs. 3 and 4).

## CONCLUSION

The study has demonstrated the significant advantages of incorporating trigonometric similarity measures within the BLDFHS framework. By leveraging cosine and cotangent functions, the proposed similarity measures provide a more refined and comprehensive way to evaluate complex decision-making scenarios. This approach excels in capturing both positive and negative aspects of membership and their respective control parameters, making it effective for multi-criteria decision-making processes.

### Outcome of the study

The mathematical properties of the trigonometric similarity measures reinforce the reliability and robustness of these measures, making them a powerful tool for practical applications. Another strength of this study is the consideration of bipolar aspects and control parameters, combined with trigonometric measures, which allows for a more detailed analysis. It enables a more intuitive interpretation of relationships between datasets. The use of cosine and cotangent functions helps assess the degree of similarity with higher sensitivity to changes in membership and non-membership attributes, which is essential for domains requiring detailed comparative analysis. Overall, the study's contribution lies in providing a comprehensive and adaptable method bridging the gap between theoretical advancements and practical applications in fuzzy decision-making models.

The proposed BLDFHS Trigonometric Similarity Measure effectively enhances the diagnostic precision for Dengue Shock Syndrome (DSS) by systematically comparing patient symptoms with established criteria. The results confirm its superiority over traditional cosine and cotangent measures, ensuring higher accuracy and reliability in clinical decision-making. This approach aids haematologists in early detection, minimizing diagnostic delays and improving patient survival rates. By addressing the complexities of uncertain and overlapping symptoms, the method provides a structured framework for timely medical intervention. The study highlights the potential of fuzzy hypersoft set-based models in advancing healthcare diagnostics and reducing DSS-related fatalities.

## Future research

Future research could focus on developing more efficient algorithms to lessen the computational demands of trigonometric similarity measures within the bipolar linear diophantine fuzzy framework. Additional studies might explore incorporating other trigonometric and hybrid similarity measures to address more complex decision-making scenarios. Expanding these measures' applications to fields like real-time decision-making, machine learning, and data analytics could be further explored to other critical medical conditions and complex real-world scenarios.

### Funding

This work was supported by the Institute of Information & Communications Technology Planning & Evaluation (IITP)-ITRC (Information Technology Research Center) grant funded by the Korea government (MSIT) (IITP-2025-RS-2024-00438335) and by INHA UNIVERSITY Research Grant. The funders had no role in study design, data collection and analysis, decision to publish, or preparation of the manuscript.

### Grant Disclosures

The following grant information was disclosed by the authors:
Institute of Information & Communications Technology Planning & Evaluation (IITP)-ITRC (Information Technology Research Center): IITP-2025-RS-2024-00438335.
INHA UNIVERSITY Research Grant.

### Competing Interests

Dragan Pamucar is an Academic Editor for PeerJ.

### Author Contributions

- J. Vimala conceived and designed the experiments, analyzed the data, prepared figures and/or tables, and approved the final draft.
- S. Nithya Sri conceived and designed the experiments, performed the experiments, performed the computation work, prepared figures and/or tables, authored or reviewed drafts of the article, and approved the final draft.
- Nasreen Kausar performed the experiments, analyzed the data, authored or reviewed drafts of the article, and approved the final draft.
- Dragan Pamucar conceived and designed the experiments, performed the computation work, authored or reviewed drafts of the article, and approved the final draft.
- Vladimir Simic analyzed the data, prepared figures and/or tables, and approved the final draft.
- Jungeun Kim conceived and designed the experiments, performed the experiments, performed the computation work, prepared figures and/or tables, and approved the final draft.

## Data Availability

Raw data is available in the Supplemental Files.

## Supplemental Information

Supplemental information for this article can be found online at http://dx.doi.org/10.7717/peerj-cs.3144#supplemental-information.

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
