# Peer review of "Medical decision support for dengue shock syndrome using a bipolar linear diophantine fuzzy hypersoft model with trigonometric similarity"

_PeerJ Computer Science, doi:10.7717/peerj-cs.3144_

## Round 0.1 · original submission · Major Revisions

While the topic is potentially valuable and relevant, the manuscript requires major revision before it can be reconsidered for publication. The authors should:

1. Thoroughly revise the manuscript for clarity, grammar, and structure, including accurate and consistent labeling of sections, examples, and references.

2. Clarify and rigorously justify the mathematical framework, ensuring all proofs and assumptions are complete and well-explained.

3. Include a comparative evaluation with existing models to validate the claimed improvements.

4. Clearly illustrate the proposed model/algorithm, supported by meaningful visual aids and explanatory text.

5. Address the relevance and interpretation of novel terms (e.g., negative membership values) within the context of real-world applications.

6. Consider expanding the case study to include real or semi-validated data, or clarify its role as a conceptual demonstration.

7. Improve the introduction and conclusion, highlighting motivation, contributions, and potential future work.

A revised version that fully addresses these issues may be considered for further review.

Reviewer 1 ·

Basic reporting

1. Some of the writing should be improved. For example, "and a two intuionistic" should be "and two intuitionistic"; the same problem is recurring in other definitions, examples, and preposition 13, etc. Also, figure and table captions are strange or unexplained. For example, Figure 1 (algorithm flowchart) doesn’t add much value because it’s too simple.

2. The references "Riaz and Hashmi (2019a)" and "Riaz and Hashmi (2019b)" seem to be referring to the same article.

3. There are some inconsistencies in how sections and examples are numbered. For example, the "Article Flow" section refers to “Section 2” for prerequisites, but that content is found in “1. Prerequisites.” The same goes for other sections as well. Also, “Example 3.7” and “Example 3.8” are labeled incorrectly—they’re Examples 11 and 12. These kinds of mistakes make it harder for readers to follow the paper and give the impression that it hasn’t been carefully edited before submitting, raising questions about the article's credibility.

Experimental design

1. Some of the key mathematical parts aren’t fully explained. For instance, in Proposition 13, the authors don’t show whether addition is well-defined (i.e., closure under operations or equivalence-class consistency). This is important because without it, the whole structure they’ve built might not work as expected.

2. In Theorem 16, they used a well-known inequality (Cauchy-Schwarz) but skipped over some important steps in the proof. The jump between lines 194 and 197 isn’t clearly explained, which makes it hard for readers to understand or verify the inequality.

Validity of the findings

1. The authors say their method is better than others, but they don’t show any clear comparisons, for example, comparisons with other established techniques. Without this, it’s hard to believe the method is better than the existing ones.

2. The paper includes negative membership values (μ_neg and ν_neg), but it doesn’t explain what they mean or why they’re useful, especially in a medical setting. If these values are part of the model, readers need to understand how they help or what they represent in real-life situations.

3. The case study uses synthetic data, and it doesn’t involve any actual medical professionals or patient data. For a paper related to healthcare, this is a big concern.

Additional comments

At this stage, I don't think the paper is ready for publication. The issues, especially related to the missing proof, unclear definitions, dummy data, and lack of comparison with well-established techniques, are too serious to overlook.

·

Basic reporting

The model proposed for handling uncertain and overlapping symptomatology significantly improves early detection. However, the proposed model should be drawn instead of the following chart.
Some questions should be addressed as follows:
- What is the advantage of the algorithm/model?
- Classify the novelty of the method.

Experimental design

The proposed model should be drawn with its explanation.

Validity of the findings

How to deal with a large number of the proposed models? Classify

Additional comments

Compassion methods should be discussed in detail since it is quite simple.

·

Basic reporting

I think it is good work for the journal, but some suggestions are listed below before going to the next round.

1. The abstract only contains some sentences without any process conditions, which is insufficient to delineate the whole picture of the contribution of this study.

2. Introduction: Sections of the manuscript are not well organized. This section can be made much more impressive by highlighting the contributions.

3. There are some grammatical errors. Please check the whole manuscript to improve the language.

4. Give a clear motivation for the paper in the introduction section.

5. Throughout my reading, I met some typos. The authors are advised to check them carefully to improve the quality of the manuscript.

6. I suggest the authors make their paper simple in content, to let readers understand the main contribution easily.

7. Add future work direction in the conclusion section.

Experimental design

-

Validity of the findings

-

---

## Round 0.2 · Minor Revisions

Authors have addressed most of the comments from the reviewers. But the following minor issues should also be clarified before this paper can be reconsidered in this journal.

1.In the "Abstract" section, the authors should include the quantitative performance results achieved by the proposed method to highlight its effectiveness.

2. In the "Contribution" and "Novelty" sections, the authors should clearly identify the key performance indicators (KPIs) in which the proposed method outperforms previously published approaches.

Reviewer 1 ·

Basic reporting

The authors have successfully addressed the concerns.

Experimental design

The authors have successfully addressed the concerns.

Validity of the findings

The authors have successfully addressed the concerns.

Additional comments

The authors have successfully addressed the concerns.

·

Basic reporting

Technical contributions should be stated clearly since the updated texts in the study contribution are so long. All texts in technical contributions should be rewritten in this paper.

Experimental design

The paper has been revised mostly based on the reviewer suggestions

Validity of the findings

The paper has been revised mostly based on the reviewer suggestions

Additional comments

The paper has been revised mostly based on the reviewer suggestions

---

## Round 0.3 · accepted · Accept

The authors have thoroughly addressed all reviewer comments, and the revisions have strengthened the manuscript. Therefore, I recommend that the paper be accepted in its current form.

·

Basic reporting

The paper has been revised completely as reviewer suggestions.

Experimental design

The paper has been revised completely as reviewer suggestions.

Validity of the findings

The paper has been revised completely as reviewer suggestions.

Additional comments

The paper has been revised completely as reviewer suggestions.